# LEARNING TO REFINE: SELF-REFINEMENT OF PARALLEL REASONING IN LLMS

## ABSTRACT

To further enhance the ability of Large Language Models (LLMs) to solve complex, multi-step reasoning problems, test-time scaling (TTS) methods have gained widespread attention. Existing approaches such as Best-of-N and majority voting are limited as their performance depends on the quality of candidate responses, making them unable to produce a correct solution when all candidates are incorrect. Introducing an additional model to select the best response also incurs significant deployment costs. To this end, we introduce Generative Self-Refinement (GSR), a novel parallel test-time scaling framework where a single and unified model first generates a set of candidate responses in parallel and then performs self-refinement to synthesize a new superior solution based on a prompt consisting of the problem and these candidates. However, LLMs struggle to perform refinement effectively when prompted directly. Therefore, we design a hybrid training pipeline by jointly optimizing for two complementary objectives, solving problems directly and refining candidate responses. Experimental results demonstrate that our method achieves state-of-the-art performance across five mathematical benchmarks. We further show that this learned self-refinement skill is a model-agnostic enhancement, robust across different model scales and generalizing to out-of-distribution reasoning tasks.

## 1 INTRODUCTION

The ascent of Large Language Models (LLMs) (Brown et al., 2020) marks a significant milestone toward Artificial General Intelligence (AGI), with the power largely stemming from training-time scaling (Kaplan et al., 2020; Hoffmann et al., 2022). Recently, a complementary paradigm has gained prominence, in which model performance can be consistently improved by allocating additional compute at inference time (Brown et al., 2024; Snell et al., 2024). This approach is formally termed **test-time scaling (TTS)**.

A key frontier in TTS is enhancing a model's ability to solve complex, multi-step problems, which often requires sophisticated strategies to push the performance boundaries of the model. A common strategy is majority voting (Wang et al., 2023), which leverages the principle of self-consistency to improve performance, identifying the most frequent answer from multiple reasoning paths. Best-of-N (BoN) approach takes this process further by introducing an external verifier, typically a Reward Model (RM), to score, rank and select the best response from candidates (Irvine et al., 2023; Song et al., 2025). Moving beyond selective mechanisms, other methods such as LLM-Blender (Jiang et al., 2023), LMCor (Vernikos et al., 2024), AFT (Li et al., 2025) and CoT-based-Synthesizer (Zhang et al., 2025a) train a dedicated model to fuse information from multiple candidate responses.

However, these prevailing strategies have fundamental limitations. The performance of majority voting and BoN, is inherently bounded by the quality of the set of candidates. They are mechanistically incapable of producing a solution that transcends the quality of candidate proposals, which becomes particularly problematic when all candidates are flawed. Moreover, this approach discards non-selected candidates entirely, losing valuable insights contained within the reasoning process. For fusion methods, training a specialized synthesizer model to fuse candidates generated by other policy models solely decouples the connection between the generation and integration, therefore precluding any synergy that might exist between these two intertwined capabilities. The design of

introducing an external model to verify or fuse imposes complex data curation, extra compute cycles and GPU memory occupancy (Ahmadian et al., 2024; Sareen et al., 2025).

To address this, we propose a novel parallel test-time scaling method, Generative Self-Refinement. The core innovation of our method is to empower a single and unified model to perform self-refinement on its own diverse parallel outputs. Concretely, our model first generates a set of candidate responses in parallel and then performs self-refinement on a prompt consisting the problem and candidates. Our model is prompted to leverage its intrinsic in-context reasoning ability and selectively synthesize valid insights from candidate responses to produce a final better solution. As illustrated in Figure 1, our model is able to identify errors in candidates and provide a correct solution even when all responses are incorrect. This finding supports a premise that even erroneous solutions discarded after selection could contain valid intermediate steps, identified pitfalls, or explored pathways. Consequently, our method achieves a significantly higher performance boundary than majority voting or Best-of-N, as it can construct a correct solution even

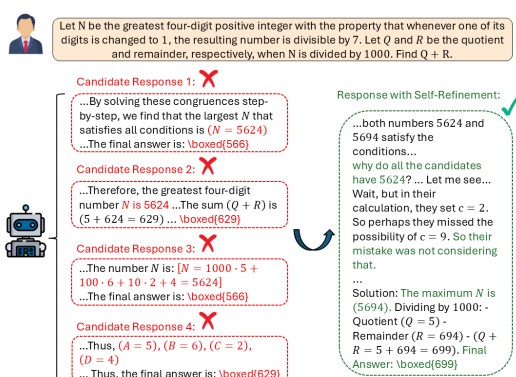

Figure 1: An example of our method. The model is provided with the original problem and a set candidate solutions generated by itself. Even provided with four incorrect candidates, the model can still reference them, diagnose the flaws, and finally construct a correct answer. We provide a more detailed case in Appendix H.

when all candidates are flawed. Empirically, our method demonstrates strong performance on problems with few correct candidate responses, which are typically the most difficult.

However, we find that advanced self-refinement capability cannot be reliably elicited through prompting alone in some cases. This is because data specifically for refining responses is absent from standard training corpora. Furthermore, we observe that there is a potential synergistic relationship between generating responses effectively and integrating multiple responses into a better answer. The effectiveness of integration at inference time is directly contingent upon the quality of the generated candidates, while training the model on the refinement task solely also in turn enhances its foundational ability for direct generation. Motivated by this, we design a hybrid training pipeline to improve the performance by jointly optimizing for two complementary objectives, direct-solving and self-refinement. This process aims to equip the model with dual abilities to generate high-quality responses directly and to self-refine existing solutions. Our key contributions are as follows:

- We propose Generative Self-Refinement, a novel test-time scaling method which can refine its own outputs to improve the performance on complex reasoning.

- To further improve the performance, we propose a hybrid training pipeline designed to train the model on complementary objectives, direct-solving and self-refinement.

- We demonstrate that our method achieves state-of-the-art results on multiple challenging mathematical benchmarks. Furthermore, extended experiments demonstrate that self-refinement is a robust, generalizable and model-agnostic skill.

## 2 RELATED WORK

**Test-time scaling** Test-time scaling (TTS) (Brown et al., 2024; Wu et al., 2025) is a promising approach for improving model's performance by increasing computation at inference time (Snell et al., 2024). TTS approaches generally fall into four categories (Zhang et al., 2025b): (1) parallel, where multiple samples are generated concurrently and then aggregated; (2) sequential, where a solution is iteratively refined; (3) hybrid, which combines both parallel generation and sequential improvement; (4) internal, which performs long-chain reasoning within the model's internal parameters. The most well-known approach within parallel scaling is majority voting (Wang et al., 2023), which leverages the self-consistency of the model's outputs to select the most consistent answer. Best-of-N (BoN) extends this principle, replacing the voting mechanism with an external verifier, which is typically

a Reward Model (RM). The external verifier is used to either assign a scalar score (Cobbe et al., 2021; Rafailov et al., 2024; Winata et al., 2025), or to perform discriminative methods to indicate a preference (Stiennon et al., 2020; Nakano et al., 2022). Some works have introduced the Generative Reward Model (GenRM) (Kim et al., 2024; Mahan et al., 2024), which leverages the model's generative capacity to provide a detailed and interpretable justification alongside the evaluation. A recent line of work (Whitehouse et al., 2025; Chen et al., 2025; Guo et al., 2025) focuses on integrating reasoning capabilities into reward modeling to enhance performance and interpretability.

**Correction Methods**   Some studies have explored that models can refine responses to improve the performance (Cobbe et al., 2021; Gou et al., 2024; Ferraz et al., 2024). Self-Refine (Madaan et al., 2023), RIC (Kim et al., 2023) and REFINER (Paul et al., 2024) focus on sequential self-refinement, where a single model is used to generate feedback on its own output and iteratively revise the solution accordingly. However, Huang et al. (2024) find that LLMs struggle to self-correct their reasoning without external feedback. Distinct from sequential approaches, LLM-Blender (Jiang et al., 2023), LMCor (Vernikos et al., 2024), AFT(Li et al., 2025), CoT-based-Synthesizer (Zhang et al., 2025a) train a dedicated generative model to fuse multiple solutions. However, these methods focus exclusively on the task of improving parallel solutions, while neglecting to train the model to solve problems directly. Besides, MoA (Wang et al., 2025) and Multiagent FT (Subramaniam et al., 2025) leverages multi-agent society of specialized LLMs to improve responses, incurring significant inference and deployment overhead.

## 3 METHODOLOGY

### 3.1 OVERVIEW

We introduce Generative Self-Refinement (GSR), a framework that generates a superior final answer by selectively leveraging insights from multiple parallel candidate solutions generated by itself. We employ a single model to first generate a set of solutions directly and then perform self-refinement to produce a superior solution after constructing the augmented prompt consisting of the problem and candidates. Our method is based on dual abilities: the ability not only to generate high-quality solutions directly but also to scrutinize and improve upon the candidate solutions. Consequently, the dual abilities offer significant flexibility during inference, enabling a choice between solving problems directly and performing self-refinement based on the trade-off between compute budget and accuracy.

### 3.2 SELF-REFINEMENT

The self-refinement process employs a single model and involves two main stages. First, the model generates a set of diverse candidate solutions in parallel. We prompt the model in a standard question-answer format. Since the thinking model reasons in a Chain-of-Thought (CoT) style, which can be verbose, we parse the raw outputs and extract only the summary content to serve as the final response. Then, these candidates are used to construct a new augmented prompt consisting of the original problem and the candidate solutions. As shown in Figure 2, we design a template to explicitly instruct the model towards a reflective and synthetic reasoning process. Since the quality of the candidate responses is unknown beforehand in practice, we explicitly inform the model of this uncertainty in the prompt. The model is prompted to first analyze the connection between the candidates and the original problem, and to selectively leverage any valuable insights from candidates. Crucially, even if all candidates are flawed, the model is required to reason independently and produce the final correct solution. The augmented prompt is subsequently passed back to the model to elicit a final improved solution. This self-refinement process is built upon the model's intrinsic reasoning ability and its ability to leverage the provided candidates.

### 3.3 HYBRID TRAINING PIPELINE

However, it is infeasible to directly prompt a model to achieve the optimal performance. To address this challenge, we design a hybrid training pipeline to endow the model with the dual abilities, generating high-quality responses as candidates and perform self-refinement. As illustrated in Figure 3, this process involves constructing a specialized dataset for the refinement task and then combing it with a traditional instruction dataset. We employ a teacher-student distillation framework. The model we aim to train is designated as the student model, while a more capable "oracle" model serves as the teacher to generate the target responses.

...provided with a challenging problem and a set of candidate responses which may be correct, partially correct or even wrong.

...You should first fully summarize the connection between the candidates and the problem ...generate a correct solution yourself if all candidates are wrong. Don't copy candidates, use insights selectively and reason independently.

Problem:

{Problem}

Candidate Response 1:

{Response 1}

Candidate Response 2:

{Response 2}

...

Figure 2: Prompt template for our method. Full prompt template is provided in Appendix D.

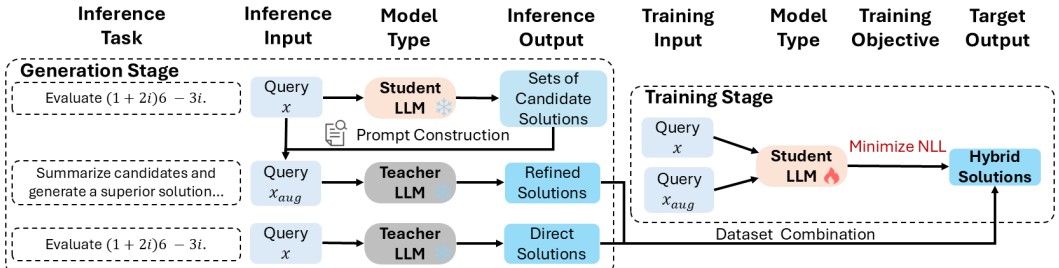

Figure 3: An overview of the hybrid training pipeline, which consists of a data generation stage (**Left**) followed by a supervised fine-tuning (SFT) stage (**Right**). We use a teacher model to construct hybrid dataset and then train the student model on dual tasks.

We now formalize the training process. Consider a dataset, $\mathcal{D} = \{q^{(i)}, a^{(i)}\}_{i=1}^{N}$, $q$ is the query and $a$ is the corresponding ground truth. Let $M_\theta$ represent the student model parameterized by $\theta$ and $P_\theta$ be the distribution over input tokens. The data generation process begins with the student model $M_\theta$ sampling $K$ diverse candidate solutions $\mathcal{O}_K = \{o_{stu}^{(1)}, \cdots o_{stu}^{(K)}\} \sim P_\theta(q, \mathcal{S})$ with its specific decoding strategy $\mathcal{S}$ for the query $q$. The objective of any TTS strategy is to maximize the expectation of obtaining the ground truth. Formally, this objective can be expressed as:

$$\max_\theta \mathbb{E}_{(q,a)\sim\mathcal{D}}[\mathbb{E}_{\mathcal{O}_K \sim P_\theta(\cdot|q)}[\mathbb{1}\{\mathcal{U}(\mathcal{O}_K) = a\}]] \tag{1}$$

where $\mathcal{U}$ denotes a TTS strategy function and $\mathbb{1}\{\cdot\}$ is the indicator function. Unlike approaches that rely on an external verifier or self-consistency, our method first concatenates the candidates $\mathcal{O}_K$ with the original query $q$ to construct an augmented prompt $q_{aug}$. In practice, we distill correct outputs of the teacher model as target to approximately optimize this objective, which makes self-refinement as a learnable inference strategy. We also sample the teacher model's direct responses to the queries as another set of training targets. Previously generated self-refinement data is combined with traditional direct-answer instruction data, to constitute the final hybrid training corpus. The training process is achieved by minimizing a composite loss function $\mathcal{L}_{distill}(\theta)$ in this hybrid dataset, which is defined as follows:

**Direct-Solving** Let $\mathcal{D}_{direct}$ be the dataset for the direct-answer task. Given a training instance $(q, o) \in D_{direct}$ where $q$ is the question and $o$ is the golden solution generated by the teacher. The objective is to maximize the conditional probability of the target sequence given the question, which can be achieved by minimizing the negative log-likelihood (NLL) loss. The loss function for a instance is defined as:

$$\mathcal{L}_{direct}(\theta; q, o) = - \sum_{t \in [|o|]} \log P_\theta(o_t | q, o_{<t}) \tag{2}$$

where $o_{<t}$ denotes the target sequence of preceding tokens $(o_1, \cdots, o_{t-1})$.

**Self-Refinement** Let $\mathcal{D}_{selfR}$ be the dataset for the self-refinement task. Given a question $q$ and $\mathcal{O}_K = \{o_{stu}^{(i)}\}_{i=1}^K$, a corresponding set of $K$ candidate solutions generated by the student model. We construct the augmented prompt $q_{aug}$ by concatenating the original question with candidates:

$$q_{aug} = q \oplus \langle sep \rangle \oplus o_{stu}^{(1)} \oplus \langle sep \rangle \oplus \cdots \oplus o_{stu}^{(K)} \tag{3}$$

where $\oplus$ denotes token sequence concatenation and $\langle sep \rangle$ denotes some separator tokens. The loss function for a single instance from dataset $\mathcal{D}_{selfR}$ is defined as:

$$\mathcal{L}_{selfR}(\theta; q, \mathcal{O}_K, o^*) = -\sum_{t \in [|o^*|]} \log P_\theta(o_t^* | q_{aug}, o_{<t}^*) \tag{4}$$

where $o^*$ is the golden solution generated by the teacher.

**Optimize Objective** The overall loss function $\mathcal{L}_{distill}(\theta)$ for model $M_\theta$ is formulated as:

$$\mathcal{L}_{distill}(\theta) = \mathbb{E}_{(q,o)\sim\mathcal{D}_{direct}}[\mathcal{L}_{direct}(\theta; q, o)] + \mathbb{E}_{(q,\mathcal{O}_K,o^*)\sim\mathcal{D}_{SelfR}}[\mathcal{L}_{selfR}(\theta; q, \mathcal{O}_K, o^*)] \tag{5}$$

## 4 EXPERIMENTS

### 4.1 EXPERIMENTAL SETUP

**Dataset Curation** We establish a data pipeline based on a single large-scale math dataset, Open-MathReasoning (Moshkov et al., 2025). After filtering and construction, we constitute the final 368K training dataset with 184K direct-answer dataset and 184K self-refinement dataset. For the self-refinement dataset, every problem consists of the original query and four candidate responses. More details are provided in Appendix F.1.

**Training Settings** We train Qwen2.5-7B-Instruct (Yang et al., 2025b) using Supervised Fine-Tuning (SFT) and employ QwQ-32B (QwenTeam, 2025) to generate target output. We refer to the fine-tuned model as GSR-7B. We present more details on parameter settings in Appendix F.2.

**Baselines** To rigorously evaluate the efficacy of our method, we compare it with the following TTS baselines: (1) **Majority Voting** (Wang et al., 2023), A non-parametric heuristic that selects the most consistent answer from a set of solutions without any external modules. (2) **Skywork-Reward-Gemma-2-27B-v0.2** (Liu et al., 2024), a state-of-the-art scalar reward model that provides a single numerical score to rank the overall quality without explanations or reasoning. (3) **RM-R1-DeepSeek-Distilled-Qwen** (Chen et al., 2025) and **RRM** (Guo et al., 2025), two concurrent approaches integrating reasoning capabilities into reward modeling, significantly surpassing conventional generative RM's performance. (4) **Synthesizer-8B-math** (Zhang et al., 2025a), which is a generative integration method that achieves SOTA performance on mathematical datasets by synthesizing and correcting multiple Chain-of-Thought (CoT) paths to produce final answers.

**Benchmarks** For a comprehensive evaluation of mathematical performance, we evaluate all baselines across five challenging and most representative benchmarks (Hochlehnert et al., 2025) in the mathematical domain: AIME24 (AI-MO, 2024a), AIME25 (Lin, 2025), AMC22 & AMC23 (AI-MO, 2024b), MATH500 (Hendrycks et al., 2021) and OlympiadBench (He et al., 2024).

**Evaluation Settings** We report direct-sovling average accuracy using pass@1. For a direct and fair comparison, we report the average metrics maj@4 (majority voting), BoN@4 (Best-of-N), cor@4 (correctoin methods), and SelfRef@4 (Generative Self-Refinement) on the same sets of four candidate responses ($k = 4$). To ensure a fair comparison of computational budget, we further report the average metrics for majority voting and non-generative BoN methods with five candidate responses ($k = 5$). We use abbreviations for Skywork-Reward-Gemma-2-27B-v0.2 (SkyRM-27B), Synthesizer-8B-math (Syn-8B), RM-R1-DeepSeek-Distilled-Qwen (RM-R1) and QwQ-32B-Preview (QwQ-Preview). More details can be found in Appendix F.3.

### 4.2 MAIN RESULTS

We present a comprehensive evaluation of our method against advanced test-time scaling baselines. The results, summarized across five challenging mathematical benchmarks, are presented in Table

Table 1: Comprehensive performance evaluation on five mathematical benchmarks. We report the results of our GSR-7B and base Qwen2.5-7B-Instruct for reference. We denote the Best-of-N and fusion methods by the name of external models used to perform them. We report the average metrics of 32 runs for AIME24 and AIME25 and 16 runs for the remaining. Additionally, we report the pass@1 metric for two leading models from Yang et al. (2025c) and QwenTeam (2024).

| Method | AIME24 | AIME25 | AMC22-23 | MATH500 | Olympiad | Average |
|--------|--------|--------|----------|---------|----------|---------|
| Larger Models | | | | | | |
| QwQ-Preview | 50.0 | 32.7 | | 90.6 | | |
| o1-mini | 56.7 | 50.8 | | 90.0 | | |
| Qwen2.5-7B-Instruct | | | | | | |
| pass@1 | 13.2 | 7.2 | 43.6 | 75.9 | 39.5 | 35.9 |
| maj@4 | 16.7 | 9.6 | 47.7 | 79.4 | 43.5 | 39.4 |
| SkyRM-27B@4 | 17.1 | 9.2 | 48.0 | 78.6 | 43.0 | 39.2 |
| selfRef@4 | 15.6 | 11.3 | 47.5 | 78.7 | 43.8 | 39.4 |
| GSR-7B | | | | | | |
| pass@1 | 50.1 | 37.8 | 78.5 | 90.6 | 64.4 | 64.3 |
| maj@4 | 60.0 | 46.7 | 84.6 | 92.8 | 68.3 | 70.5 |
| maj@5 | 60.7 | 48.3 | 84.9 | 93.1 | 68.8 | 71.1 |
| Syn-8B | 60.2 | 44.0 | 84.0 | 92.0 | 67.7 | 69.6 |
| SkyRM-27B@4 | 58.3 | 46.7 | 81.6 | 92.5 | 67.6 | 69.3 |
| SkyRM-27B@5 | 59.2 | 46.3 | 82.8 | 92.7 | 67.7 | 69.7 |
| RRM-7B | 60.4 | 46.3 | 80.4 | 90.6 | 64.6 | 68.5 |
| RM-R1-7B | 62.1 | 45.4 | 84.6 | **93.5** | 69.1 | 70.9 |
| selfRef@4 | **66.0** | **51.7** | **85.7** | 93.4 | **71.0** | **73.6** |

1. A primary observation is the substantial performance uplift after the training process. Compared to the base model, GSR-7B exhibits dramatic improvements across all standard metrics, particularly on the most challenging benchmarks. For instance, its pass@1 accuracy on AIME24 increases from a modest 13.2% to 50.1%, confirming the efficacy of our approach.

Our method shows significant improvement after post-training. Before training, the performance of selfRef@4 on the base model is comparable with majority voting. After training, our method emerges as the **state-of-the-art** method, achieving 73.6% average accuracy. It surpasses not only standard baselines such as majority voting but also more complex Best-of-N (BoN@4), as demonstrated on AIME24 (66.0% vs. 62.1%). We note that in the MATH500 dataset, selfRef@4 is marginally underperformed by BoN@4 (93.4% vs. 93.5%). We hypothesize that it is due to the specific dynamics of high-accuracy regimes (90.6% on pass@1). In such a scenario, at least a correct answer is highly likely to be present in any set of candidate responses, making selection methods like Best-of-N particularly effective.

Overall, the experimental results validate the effectiveness of the model's dual abilities. GSR-7B's pass@1 accuracy (e.g., 50.1% on AIME24 and 37.8% on AIME25) makes it competitive with the QWQ-32B-Preview. With the addition of our parallel test-time scaling method, it can even surpass OpenAI o1-mini, as shown by AIME24 (66.0% vs. 56.7%) and AIME25 (51.7% vs 50.8%).

### 4.3 FINE-GRAINED ANALYSIS ON SELF-REFINEMENT

Although the overall accuracy metrics are informative in Section 4.2, the underlying mechanism of our approach is still obscure. To dissect the true capabilities of our method, we conduct a more fine-grained conditional analysis. The goal is to understand how our method performs when provided with imperfect or even entirely incorrect candidates. We report the average accurate rates conditioned on the number of correct candidates out of 4 ($N_c \in \{0, 1, 2, 3, 4\}$).

**Comparison with Baselines** As shown in Table 2, we first compare various parallel test-time scaling approaches in AIME24 benchmark. When a clear majority of candidates are correct ($N_c \geq 3$),

all methods perform exceptionally well, often achieving near-perfect accuracy. However, our methods begin to differentiate in more ambiguous cases ($N_c < 3$). When only one candidate is correct, our selfRef@4 achieves an impressive 60.2% accuracy, substantially outperforming majority voting (maj@4 at 18.8%), cor@4 (38.3%) and all BoN@4 variants. The most illuminating results, even when all candidates are incorrect ($N_c = 0$), our approach can still produce a correct solution 5.9% of the time, whereas all baseline methods fail completely. These low $N_c$ scenarios serve as a proxy for a class of most challenging problems where generating a correct solution is difficult. The effectiveness of our method in these situations highlights its robustness for solving difficult problems. In contrast, both majority voting and best-of-N are incapable of producing a correct answer if no correct candidate exists. The correction method (syn-8B) also falters on more challenging benchmarks. We hypothesize that training exclusively on synthesis while neglecting the continued training of foundational reasoning, which in turn acts as a bottleneck, constrains the effective application of the synthesizing faced with more complex problems.

**Generalization across Benchmarks**    To verify that the superiority of our approach is not dataset-specific, we report the performance across five diverse benchmarks in Table 3. The findings reveal a remarkable consistent trend. Across all benchmarks, our model's ability to recover from a complete set of incorrect candidates ($N_c = 0$) is consistently demonstrated, ranging from 3.4% on MATH500 to 9.0% on AMC22-23. This consistency proves this capability is not a fluke but a generalizable skill that our model has acquired.

Table 2: Fine-grained analysis on the AIME24. We report accurate rates in percentage (%) conditioned on the number of correct candidates ($N_c$).

| Method | $N_c$ | | | | |
| --- | --- | --- | --- | --- | --- |
| | 4 | 3 | 2 | 1 | 0 |
| maj@4 | 100 | **100** | **94.4** | 18.8 | 0.0 |
| Syn-8B | 99.6 | 96.2 | 83.3 | 38.3 | 0.0 |
| SkyRM-27B | 100 | 79.5 | 83.3 | 43.8 | 0.0 |
| RM-R1-7B | 100 | 94.9 | 80.6 | 56.3 | 0.0 |
| RRM-7B | 100 | 89.7 | 77.8 | 53.1 | 0.0 |
| selfRef@4 | 100 | 97.4 | 89.6 | **60.2** | **5.9** |

Table 3: Robustness analysis of our method across five diverse benchmarks. The full details are provided in Appendix Table 9.

| Benchmark | $N_c$ | | | | |
| --- | --- | --- | --- | --- | --- |
| | 4 | 3 | 2 | 1 | 0 |
| AIME24 | 100 | 97.4 | 89.6 | 60.2 | 5.9 |
| AIME25 | 100 | 95.8 | 87.1 | 65.6 | 4.5 |
| AMC22-23 | 99.9 | 99.3 | 80.0 | 53.6 | 9.0 |
| MATH500 | 99.8 | 94.3 | 76.4 | 49.1 | 3.4 |
| Olympiad | 99.8 | 94.0 | 81.0 | 50.7 | 4.4 |

## 4.4    ABLATION STUDY ON TRAINING

To validate our claim that a hybrid training strategy is essential for both direct-solving and self-refinement capabilities, we conduct a ablation study. To ensure a fair comparison, we construct three distinct training datasets randomly sampled from the total dataset, all of which contain exactly 20,000 samples and share an identical set of questions: (1) Direct-Solving Only: 20k samples from $\mathcal{D}_{direct}$. (2) Refinement Only: 20k samples from $\mathcal{D}_{selfR}$. (3) Hybrid: a balanced mix of 10k from $\mathcal{D}_{direct}$ and 10k from $\mathcal{D}_{selfR}$.

Table 4: Ablation study on different training strategies. We compare hybrid training against strategies using only direct-solving data (direct) or only refinement data (ref).

| Method | Benchmark | Base Model | SFT (20k direct) | SFT (20k ref) | SFT (20k hybrid) |
| --- | --- | --- | --- | --- | --- |
| pass@1 | AIME24 | 13.2 | **27.9** | 26.4 | 27.5 |
| | AIME25 | 7.2 | **26.7** | 23.3 | 25.6 |
| selfRef@4 | AIME24 | 15.6 | 37.5 | 45.0 | **45.6** |
| | AIME25 | 9.6 | 30.2 | 30.0 | **32.7** |

The results presented in Table 4 lead to a clear conclusion. The model trained on a Direct-Solving Only dataset achieves the highest pass@1 scores. However, its capacity for self-refinement is severely limited, lagging significantly behind hybrid model. It demonstrates that generative self-refinement is not an innate or emergent property but a skill that must be explicitly learned.

To our surprise, even with training purely on the Refinement Only dataset, the model still achieves a higher pass@1 score than the base model. This indicates that learning the skill of refinement

imparts a more general reasoning ability which directly benefits the model's foundational reasoning performance. However, this model excels at the self-refinement task but demonstrates subpar pass@1 performance compared with two others. This reveals that the efficacy of the self-refinement at inference is contingent upon the model's ability to first generate strong candidate solutions.

Hybrid model strikes the optimal balance. It achieves the highest selfRef@4 scores across both benchmarks while maintaining a highly competitive pass@1 performance, confirming the necessity of the hybrid training pipeline.

## 4.5 ROBUSTNESS AND GENERALIZATION ANALYSIS

To further investigate the robustness and underlying mechanism of our framework, we conduct additional experiments on two larger-scale models: Qwen2.5-14B-Instruct and Qwen2.5-32B-Instruct (Yang et al., 2025b). This investigation is designed to address two critical questions: (**RQ1**) Is our method effective across different model scales? (**RQ2**) Does the model learn a generalizable improvement heuristic, or does it merely learn to correct its own specific errors? We use the same hybrid training dataset and experimental setup as described in Section 4.4.

Table 5: Performance on AIME benchmarks across different model scales. The table compares two fine-tuned models (w SFT) against their base counterparts (w/o SFT).

| Settings | Method | Qwen2.5-14B-Instruct | | Qwen2.5-32B-Instruct | |
| --- | --- | --- | --- | --- | --- |
| | | AIME24 | AIME25 | AIME24 | AIME25 |
| w/o SFT | pass@1 | 13.5 | 12.7 | 16.7 | 12.9 |
| | maj@4 | 17.5 | 15.8 | 19.2 | 15.8 |
| | selfRef@4 | 16.0 | 15.8 | 22.3 | 16.3 |
| w SFT | pass@1 | 49.4 | 36.9 | 66.3 | 54.2 |
| | maj@4 | 56.7 | 41.7 | 72.5 | 59.2 |
| | selfRef@4 | **68.1** | **49.6** | **75.2** | **67.3** |

**Robustness Across Different Model Sizes**  The results, presented in Table 5, demonstrate the effective scalability of our approach to models of a larger-scale. After training, both the 14B and 32B models exhibit substantial performance improvements across standard metrics (pass@1 and maj@4) compared to their w/o SFT counterparts. More critically, the results demonstrate the effectiveness of our method. For fine-tuned models, selfRef@4 consistently outperforms their respective majority voting baselines. For instance, the fine-tuned 14B model achieves a 20.1% relative improvement (68.1% vs. 56.7%) in AIME24, while the 32B model shows a 13.7% relative improvement (67.3% vs. 59.2%) in AIME25. This phenomenon contrasts with the 14B base model, which fails to benefit from self-refinement and even shows a performance degradation in AIME24 compared to the majority voting baseline. To our surprise, the base 32B model already exhibits a slight advantage with selfRef@4 over maj@4. We hypothesize that this may be due to the more advanced in-context learning (ICL) capability of the 32B model relative to smaller models. This finding justifies the use of larger powerful reasoning model as the teacher in our training pipeline.

**Generalization via Data Decoupling**  A crucial aspect of this experiment is that we reuse the hybrid dataset from Section 4.4 to train larger models. In particular, all candidate solutions within this dataset are generated by Qwen2.5-7B-Instruct. Although there is a mismatch between the candidate generator and learners, the results clearly indicate that both models successfully acquire the ability of self-refinement. This finding provides compelling evidence for a decoupling of the model's internal knowledge representation from the self-refinement skill. It demonstrates that the training process does not simply teach a model to patch its own flaws. Rather, it fosters a general, model-agnostic ability to evaluate proposed solutions regardless of their origin and aggregate a superior one.

## 4.6 INPUT SCALING ANALYSIS

To scale up to a larger number of inputs, previous work (Zhang et al., 2025a; Guo et al., 2025) has often employed a hierarchical strategy, partitioning candidate responses into fixed-size groups, producing outputs for each group, and then iteratively combining across groups. This approach has

been shown to be general and effective. However, such methods are not indeed scaling on input and have a time complexity of $\mathcal{O}(N)$.

Our experiments are conducted on a fixed set of four candidate responses. To evaluate our model's extrapolation capability, we conduct an input scaling experiment. Specifically, we evaluate performance with the number of candidate responses, $k$, ranging from 2 to 16. We compare our method with the maj@k (majority voting) and oracle pass@k. In Figure 4, for a smaller number of candidates ($k \leq 4$), our method shows strong performance, even slightly surpassing the pass@2 (63.2% vs. 62.5%) on AIME24 at $k = 2$. For a larger number of candidates ($k \leq 10$), our method consistently outperforms majority voting, and its performance scales with the number of candidates. This suggests that our method effectively extrapolates to a number of candidates greater than four. However, as the number of candidates is further increased, the performance of our method begins to saturate and even shows a slight decline. We hypothesize that this is due to the increase in the input, which increases additional noise and complexity, thereby disrupting the model's attention mechanism.

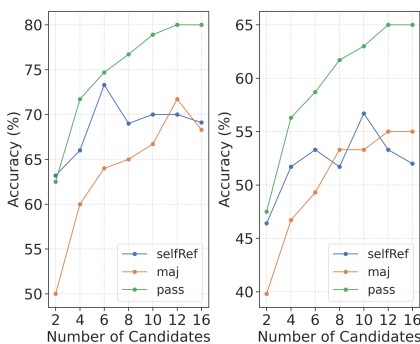

Figure 4: Accuracy of input scaling on the performance on the (**Left**) AIME24 and (**Right**) AIME25 benchmarks.

### 4.7 OUT-OF-DISTRIBUTION ANALYSIS

GSR-7B is trained exclusively on math datasets. To further analyze its generalization ability to an unseen domain, we test it on the Knights and Knaves (K&K) logic puzzles dataset (Xie et al., 2025), where some characters tell the truth and others only lie. For evaluation, we use the 4ppl subset of the K&K dataset and reformat original problems as multiple-choice questions to allow more accurate scoring. We report the performance of pass@1, majority voting (maj@4) and our method (selfRef@4). In Figure 5, GSR-7B demonstrates a substantial performance improvement over the base model. Compared to the base model's pass@1 accuracy of 12.9%, our model achieves 37.3%. More critically, we observe a clear divergence in self-refinement ability. The base model's selfRef@4 performance decreases by 7.6% relative to maj@4, while our model shows a performance improvement of 7.2%. These results confirm that the self-refinement ability is a skill acquired during training and that this skill even successfully generalizes to an unseen out-of-distribution domain.

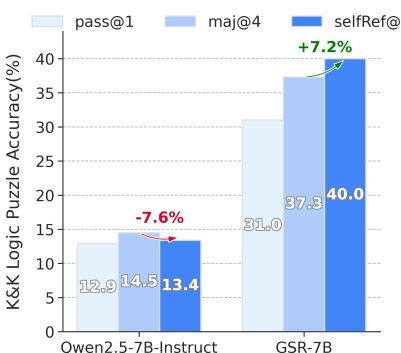

Figure 5: Experiments results for base model (**Left**) and our model (**Right**) on a subset of K&K dataset. The arrows and percentages quantify the relative performance change when applying our method over majority voting.

## 5 CONCLUSION

In this paper, we propose Generative Self-Refinement, a novel parallel test-time scaling method by which a single and unified model performs self-improvement on its own parallel solutions. This is achieved through a training process that optimizes for a hybrid objective. Our extensive experiments demonstrate that our approach consistently outperforms strong baselines. Even in the most challenging scenarios, where all candidate responses are incorrect, our model is still able to produce a correct solution. Furthermore, we find that this skill is generalizable to out-of-distribution reasoning tasks, applicable across various model scales, and decoupled from the model's specific internal parameters. This indicates that we have established a universal and model-agnostic reasoning methodology. Finally, our self-contained refinement framework highlights a promising direction for developing more capable and efficient LLM reasoners.

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

## A    USE OF LLMS

We only use Large Language Models (LLMs) to polish the manuscript and correct grammatical errors. The research content, including all ideas, experimental design and findings, is not cooperated with LLMs.

## B    ETHICS STATEMENT

We acknowledge the ICLR Code of Ethics and confirm that our work adheres to its principles. Our research is conducted on the OpenMathReasoning dataset (Moshkov et al., 2025), which is a publicly available collection of mathematical problems curated from community forums and has undergone benchmark decontamination. The dataset does not contain personally identifiable information or other sensitive data. Our study does not involve human subjects.

## C    REPRODUCIBILITY STATEMENT

We are committed to ensuring the reproducibility of our research. Our experiments are based on the publicly available OpenMathReasoning dataset (Moshkov et al., 2025). The details about our data curation is provided in Appendix F.1. The training settings, including all hyperparameters, are documented in Appendix F.2. Our evaluation settings are thoroughly detailed in Appendix F.3.

To further foster replication of our results, we provide our source code as part of the supplementary materials.

## D    PROMPT TEMPLATE

Our model support dual abilities during inference: direct-answer generation and self-refinement. The choice between modes is achieved by constructing different input prompts. Here we provide the detailed prompt template used to prompt our model to perform self-refinement. The prompt template consists of the original problem and a set of candidate solutions mainly.

---

**Prompt Template on the Self-Refinement Mode for Generative Self-Refinement**

You are an expert and creative solver, provided with a challenging problem and a set of candidate responses which may be correct, partially correct or even wrong.
You should first fully summarize the connection between candidate responses and problem, then generate a new and superior solution. You should generate a correct solution yourself if all candidates are wrong. Don't copy candidates, use insights selectively and reason independently.
Problem:
{Problem}
Think step by step and put final answer within \boxed{}.

Candidate Response 1:
{Response 1}

Candidate Response 2:
{Response 2}

Candidate Response 3:
{Response 3}

Candidate Response 4:
{Response 4}

---

We also provide a common prompt template to instruct the model to solve the problem directly. This common prompt template is applicable for any LLMs.

> **Common Prompt Template for Direct-Answer Generation**
>
> {Problem}
> Think step by step and put final answer within \boxed{}.

# E    ANALYSIS OF CANDIDATE INPUTS BURDEN

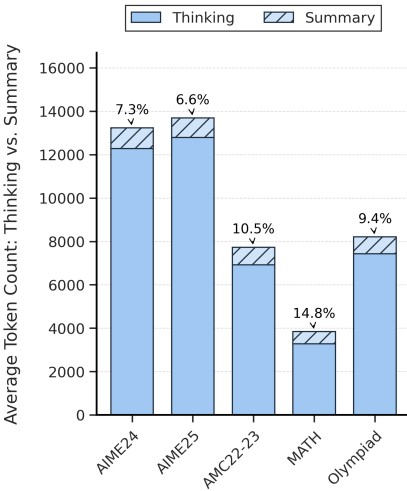

Figure 6: Average token counts of model **Direct-Solving** outputs across five benchmarks, showing the breakdown between the Thinking (solid) and Summary (hatched) components. Percentages indicate the proportion of summary tokens.

A critical factor for the feasibility of our method is the management of context length, as recontextualizing the verbose responses from thinking model can lead to prohibitive compute overhead. We innovatively mitigate this by exclusively extracting the summary component from responses while discarding the preceding thinking component.

To quantify the efficiency of this strategy, we analyze the token composition of our model's outputs across the five benchmarks of Section 4.2. As illustrated in Figure 6, the summary component is remarkably succinct, consistently accounting for only a small fraction of the total generated tokens (e.g., 6.6% to 14.8%). In particular, for difficult benchmarks like AIME24 and AIME25, the total number of tokens generated is substantial and exceeds 13,000 on average. Despite this, the summary component itself still remains remarkably concise, with an average token count of fewer than 1,000. Across the five distinct and representative benchmarks, the token length of the summary component varies from 570 to 960 tokens. This demonstrates that the input context is well controlled, imposing an average input burden of less than 4000 tokens even on the challenging AIME24 benchmark.

As detailed in Table 6, we further calculate the average token length for GSR-7B to perform self-refinement given four candidate responses. We find that token consumption is significantly lower than the direct-solving approach. We attribute this efficiency to the model's ability to leverage valuable information from correct candidate responses and thus to prune the search space significantly during its chain-of-thought reasoning.

Table 6: Average token counts of model **Self-Refinement** outputs across five benchmarks.

| Benchmark | AIME24 | AIME25 | AMC22-23 | MATH500 | Olympiad |
|---|---|---|---|---|---|
| **Token Length** | 8878 | 10477 | 5270 | 2745 | 5910 |

# F  EXPERIMENTS DETAILS

## F.1  DATASET CURATION

For simplicity, we establish a data pipeline based on a single large-scale mathematics data set, Open-MathReasoning(Moshkov et al., 2025). OpenMathReasoning dataset is curated from the AoPS community forums, performed rigorous filtering, classification, transformation, and benchmark decontamination, containing 540K mathematical problems and 3.2M generations from DeepSeek-R1 and QwQ-32B.

The detailed procedure, including the numbers of problems and samples remaining at each stage, is documented in Table 7. In summary, the pipeline involves several filtering steps, such as removing generations not used in the Kaggle competition, removing problems without an extracted answer, and discarding any generations without a pass rate evaluated by Qwen2.5-72B-Instruct. Subsequently, we aggregate the corresponding solutions for each problem, and then retaining problems with a pass rate between 0.25 and 0.9, yielding a prefiltered pool of 56K unique problems and 617K generations.

Table 7: Step by step filtering process for the construction of problems. The process starts from a large-scale raw dataset and applies a series of filtering operations to yield the final curated dataset. The number of unique problems and their corresponding generated outputs are tracked at each step.

| Step | Filtering Operation | Problems | Generations |
|---|---|---|---|
| - | Initial Raw Dataset | 540K | 3.2M |
| 1 | Remove generations not used in Kaggle competition. | - | 2.2M |
| 2 | Retain data with 'problem_type' == 'has_answer_extracted'. | - | 1.3M |
| 3 | Discard problems that have no 'pass_rate_72b_tir'. | - | 1.2M |
| 4 | Aggregate generations by their corresponding unique problem. | 116K | 1.2M |
| 5 | Filter problems to keep only those with a 'pass_rate_72b_tir' between 0.25 and 0.90. | 56K | 617K |

We construct the self-refinement dataset ($\mathcal{D}_{selfR}$) using theses 56K problems. As summarized in Table 8, for each problem, we generate 6 candidate responses with Qwen2.5-7B-Instruct using temperature of 1.0, top-p of 0.95, and a maximum output length of 4096 tokens to foster diversity. Given the inherent difficulty of the problems, we implement a specific selection process for the 6 responses. We filter and construct a fixed size set of 4 candidate responses. This set is composed of all correct solutions from the initial candidate pool of 6, with the remaining filled by incorrect solutions to meet the required size of 4. The order of these 4 candidates is then randomized. By combining each original problem with its set of candidate responses according to the prompt template in Appendix D, we constructed a dataset of 56K self-refinement problems. We further remove data where the prompt length exceeded 8,192 tokens, resulting in a final dataset of 53K samples.

Table 8: Step by step curation of final hybrid training datasets from the prefiltered pool.

| Path | Step | Operation | Problems | Generations |
|---|---|---|---|---|
| - | - | Prefiltered Pool (from Table 7 Step 5). | 56K | 617K |
| $\mathcal{D}_{selfR}$ | 1 | Use all 56K problems for self-refinement generation. | 56K | - |
| | 2 | Generate 6 solutions as candidates using Qwen2.5-7B-Instruct. | 56K | 336K |
| | 3 | Construct augmented prompts for self-refinement. | 56K | - |
| | 4 | Remove data where the prompt length exceeded 8,192 tokens. | 53K | - |
| | 5 | Generate 10 solutions for per problem using QwQ-32B. | 53K | 530K |
| | 6 | Subsample final solutions based on correctness. | - | 184K |
| $\mathcal{D}_{direct}$ | 1 | Randomly sample 184K generations from the pre-filtered pool. | - | 184K |
| Final Dataset | 1 | Merge the direct-answer dataset with self-refinement dataset. | - | 368K |

For these 53K self-refinement problems, we utilize QwQ-32B model and generate up to 10 solutions for each problem in our dataset. We use temperature of 0.7, top-p of 0.95, and limit generations to 16,384 tokens. The generated solutions by QwQ-32B are then filtered based only on the correctness. Specifically, if a problem yields between one and nine correct solutions (out of 10), we retain all of them. However, for problems where all 10 generated solutions are correct, we randomly sample four of them. This curation process yields our final training dataset $\mathcal{D}_{selfR}$ for self-refinement task of 184K samples.

For the direct-answer dataset, we randomly sample an equal number of 184K, the size of our self-refinement dataset, from the dataset obtained after Step 5 in Table 7 to create our $\mathcal{D}_{direct}$ dataset. We then merge the self-refinement dataset $\mathcal{D}_{selfR}$ with direct-answer dataset $\mathcal{D}_{direct}$ to create our final hybrid 368K training dataset.

## F.2 Training Settings

We train Qwen2.5-7B-Instruct(Yang et al., 2025b) on the 368K SFT dataset for 3 epochs with the AdamW optimizer, employing a 10% linear warmup followed by a cosine learning rate decay schedule. The maximum learning rate is set to 1e-4, with a batch size of 128 and a maximum sequence length of 24K tokens. To support longer context windows and align with advanced thinking mode, we adopted the chat_template from the Qwen3 family(Yang et al., 2025a) and extended the maximum sequence length by setting max_position_embeddings to 65,536.

In the ablation study, we train the model for 5 epochs and set the maximum learning rate to 4e-5. All other settings remain the same.

## F.3 Evaluation Settings

We explicitly instruct all models to think step by step and to enclose the final answer within \boxed{} (Hochlehnert et al., 2025). We use the Math-Verify framework, a more robust way to extract and verify answer.

We first sample candidate responses in the direct-solving mode and then perform various test-time scaling strategies based on these candidate responses. For base Qwen2.5-7B-Instruct, we configure the evaluation process to set the maximum new tokens of 4,096 and apply optimal hyperparameters. For our GSR-7B, we configure the maximum new tokens of 32,768, temerature of 0.6 and top-p 0.95. To mitigate the evaluation variance (Hochlehnert et al., 2025), we repeat 32 trials for every problem in AIME24 and AIME25, and 16 for all other benchmarks. We present the average accuracy across 16 or 32 samples generated directly for every problem as pass@1.

Then, we compute several TTS metrics based on non-overlapping groups of every four candidate responses: maj@4 (majority voting accuracy), BoN@4 (Best-of-N), cor@4 (correction methods), and SelfRef@4. The final scores for these metrics are the average performance across all groups. For all baseline models, we follow the exact hyperparameter values and the specific prompt recommended in their official documentation or model cards. For RRM and RM-R1, we employ a knockout tournament strategy (Guo et al., 2025), a method of iterative pairwise comparison and elimination to determine the best answer, which effectively guides LLMs to perform BoN sampling. Our method is evaluated with a maximum of 32,768 output tokens, 6,144 tokens for candidate response input (1,566 tokens per candidate response), temperature of 0.6 and top-p 0.95. We ensure that our method also runs 32 trails for AIME24 and AIME25, and 16 for the remaining benchmarks.

## G  Full Results of Fine-Grained Analysis

In this section, we provide the full experiment results in Section 4.3. The results of fine-grained analysis of our method across AIME24, AIME25, AMC22-23, MATH500, Olymiad is provided in Table 9. For each number of correct candidate responses (from 0 to 4), we compile statistics on our model's performance, including the number of correct answers, wrong answers, total trials, and the final correct ratio.

## H  Case Study

In this section, we present a more detailed case study from AIME24. This case demonstrates that our method can recover from four incorrect candidate responses and produce a final correct solution.

### H.1 Original Question and Candidate Solutions

The following is the original problem and four candidate responses provided to the model. Due to space constraints, the content of the candidate responses has been condensed.

Table 9: Full results of robustness analysis of our method across five mathematical benchmarks. We report correct number, wrong number, correct ratio and total conditioned on the number of correct candidates.

| Number of Correct Candidates | Statistics | Benchmarks | | | | |
|---|---|---|---|---|---|---|
| | | AIME24 | AIME25 | AMC22-23 | MAHT500 | Olympiad |
| 4 | Correct | 260 | 200 | 867 | 6604 | 5573 |
| | Wrong | 0 | 0 | 1 | 12 | 11 |
| | Correct Ratio(%) | 100 | 100 | 99.9 | 99.8 | 99.8 |
| | Total | 260 | 200 | 868 | 6616 | 5584 |
| 3 | Correct | 152 | 92 | 147 | 581 | 1049 |
| | Wrong | 4 | 4 | 1 | 35 | 67 |
| | Correct Ratio(%) | 97.4 | 95.8 | 99.3 | 94.3 | 94.0 |
| | Total | 156 | 96 | 148 | 616 | 1116 |
| 2 | Correct | 129 | 101 | 80 | 165 | 564 |
| | Wrong | 15 | 15 | 20 | 51 | 132 |
| | Correct Ratio(%) | 89.6 | 87.1 | 80 | 76.4 | 81.0 |
| | Total | 144 | 116 | 100 | 216 | 696 |
| 1 | Correct | 77 | 84 | 30 | 112 | 367 |
| | Wrong | 51 | 44 | 26 | 116 | 357 |
| | Correct Ratio(%) | 60.2 | 65.6 | 53.6 | 49.1 | 50.7 |
| | Total | 128 | 128 | 56 | 228 | 724 |
| 0 | Correct | 16 | 19 | 14 | 11 | 117 |
| | Wrong | 256 | 401 | 142 | 313 | 2563 |
| | Correct Ratio(%) | 5.9 | 4.5 | 9.0 | 3.4 | 4.4 |
| | Total | 272 | 420 | 156 | 324 | 2680 |

---

**Question and Candidate Solutions**

You are an expert and creative solver, provided with a challenging problem and a set of candidate responses which may be correct, partially correct or even wrong.

You should first fully summarize the connection between candidate responses and problem, then generate a new and superior solution. You should generate a correct solution yourself if all candidates are wrong. Don't copy candidates, use insights selectively and reason independently.

Problem:

Let $\triangle ABC$ have circumcenter $O$ and incenter $I$ with $\overline{IA} \perp \overline{OI}$, circumradius 13, and inradius 6. Find $AB \cdot AC$. Let's think step by step and output the final answer within \boxed{}. Candidate Response 1:

...Given the symmetry and the conditions, we can use the known result for the product of the sides in a right triangle:

$$AB \cdot AC = 384$$

Thus, the product $AB \cdot AC$ is:

$$\boxed{384}$$

Candidate Response 2:
...Thus, the final answer is

$$\boxed{26\sqrt{39}}$$

Candidate Response 3:
...Thus, the final answer is:

$$\boxed{312}$$

Candidate Response 4:
...Thus, the final answer is:

$$\boxed{26}$$

## H.2 Thinking Process of Self-Refinement

In this section, we present a more detailed trace of the model's thinking component and summary component, when provided with a problem and four candidate responses. We split and structure the model's raw thinking process into four parts, each marked by **a bold title**. To provide analytical insights, we have inserted *meta comment in blue italics*, focusing on how the model analyzes, discerns, and selectively utilizes valuable information from four candidates responses. We give special attention to the model's ability to self-correct by analyzing candidate responses, identifying flaws and choosing an alternative decoding path. To make the model's direct engagement explicit, its verbatim references to the candidate responses are highlighted in red.

---

**Thinking component of GSR-7B**

**1. Initial analysis and calculation (Consistent with candidates)**
Okay, let me try to tackle this problem step by step. ... First, I remember that there's a formula connecting the distance between the incenter and circumcenter ($OI$) with the circumradius ($R$) and inradius ($r$). The formula is $OI^2 = R(R - 2r)$. Let me plug in the given values: $R = 13$ and $r = 6$. So, $OI^2 = 13(13 - 12) = 13 \times 1 = 13$, so $OI = \sqrt{13}$. That's straightforward. Next, the problem states that $IA$ is perpendicular to $OI$. Since $IA \perp OI$, triangle $OIA$ is a right triangle with right angle at $I$. The hypotenuse here is $OA$, which is the circumradius, so $OA = 13$. Using the Pythagorean theorem on triangle $OIA$: $OA^2 = OI^2 + IA^2$. Plugging in the known values: $13^2 = (\sqrt{13})^2 + IA^2 \rightarrow 169 = 13 + IA^2 \rightarrow IA^2 = 156 \rightarrow IA = \sqrt{156}$.

*[Meta comment: The model's initial calculations align perfectly with the correct steps found in Candidates 1 and 2, thereby establishing a valid foundational starting point. The model also notes a critical flaw in Candidates 3 and 4: they employ an incorrect formula, $OI^2 = R^2 - 2Rr$. Although this formula coincidentally produces the correct numerical value for the specific given radius $R = 13$, its structural inaccuracy leads the model to correctly flag these candidates as unreliable early in the process.]*

**2. Deep dive and critique of candidate strategies**
*[Meta comment on Candidate 1: The model observes that Candidate 1's final answer, 384, is presented without a clear derivation. The model formulates a hypothesis: this answer might originate from an unstated and potentially incorrect assumption that the triangle is right-angled at vertex A. It then proceeds to rigorously test this hypothesis.]*

Alternatively, perhaps the triangle is right-angled? If $\triangle ABC$ is right-angled, then the circumradius would be half the hypotenuse. Since the circumradius is 13, then the hypotenuse would be 26. Suppose angle A is $90°$, then BC would be the hypotenuse, so $BC = 26$. Then, using the inradius formula: $r = (AB + AC - BC)/2$. Given $r = 6$, so $(AB+AC-26)/2 = 6 \rightarrow AB+AC = 38$. Also, by Pythagoras, $AB^2+AC^2 = 26^2 = 676$. ... $2AB \cdot AC = 1444 - 676 = 768 \rightarrow AB \cdot AC = 384$. Wait, that's one of the candidate answers (Candidate 1's answer is 384). But does this satisfy the condition $IA \perp OI$? ... Let me set coordinates: Let's place $A$ at $(0, 0)$, $B$ at $(0, b)$, $C$ at $(c, 0)$. ... in a right-angled triangle at A(0,0), with legs on axes, ... the incenter is at $(r, r)$. So here, I would be at $(6, 6)$. The circumcenter $O$ is the midpoint of $BC$, which is at $(c/2, b/2)$. Now, we need to check if $IA$ is perpendicular to $OI$. Vector $\vec{IA}$ is from I(6,6) to A(0,0): $(-6, -6)$. Vector $\vec{OI}$ is from O(c/2, b/2) to I(6,6): $(6 - c/2, 6 - b/2)$. Their dot product should be zero: $(-6)(6-c/2)+(-6)(6-b/2) = 0 \rightarrow -36+3c/2-36+3b/2 = 0 \rightarrow -72+(3/2)(b+c) = 0$. But since $b + c = 38, \rightarrow -72 + (3/2) * 38 = -72 + 57 = -15 \neq 0$. Thus, the dot product is not zero, so the condition $IA \perp OI$ is not satisfied. ...but the calculation gave AB·AC=384, which is the first candidate's answer, but it doesn't satisfy the perpendicular condition. So that must be incorrect... Therefore, the triangle cannot be right-angled at A. Hmm, so that approach was wrong.

*[Meta comment on Candidate 2: The model shifts its analysis to the other candidates and identifies a valuable, correct intermediate step within Candidate 2's reasoning. It decides to*

---

*pursue this line of reasoning to evaluate its potential and identify where, if at all, it deviates from a correct solution path.]*

...Wait, another formula for $IA$ is $IA = 2R\sin(A/2)$. Let me check that. Yes, I think that's correct. ... $IA = 2R\sin(A/2) \rightarrow \sqrt{156} = 2 \cdot 13 \cdot \sin(A/2) \rightarrow \sin(A/2) = \sqrt{156}/(26) = \sqrt{39}/13$. ... Thus, $\sin A = 2\sin(A/2)\cos(A/2) = 2 \cdot (\sqrt{39}/13) \cdot (\sqrt{130}/13) = \cdots = (2\sqrt{30})/13$. ... We had $\sin A = 2\sqrt{30}/13$. Then $\cos A = \sqrt{1 - \sin^2 A} = \sqrt{1 - (4 \cdot 30)/169} = \sqrt{(169 - 120)/169} = \sqrt{49/169} = 7/13$.

*[Meta comment: The model confirms the validity of Candidate 2's calculations regarding the trigonometric properties of angle A, acknowledging this as a key insight. However, it also diagnoses the critical failure point in Candidate 2's approach: the strategy becomes untenable when it hinges on calculating the semiperimeter, 's', a step where the candidate's method falters. The model elects to adopt the correct trigonometric values derived by Candidate 2 while consciously avoiding the flawed semiperimeter-dependent strategy.]*

...So, $6s = (1/2) \cdot AB \cdot AC \cdot (2\sqrt{30}/13) \rightarrow 6s = (AB \cdot AC \cdot \sqrt{30})/13 \rightarrow AB \cdot AC = (6s \cdot 13)/\sqrt{30}$. Hmm, so if I can find $s$, then I can compute $AB \cdot AC$... This is the path Candidate 2 took, but they failed to find $s$ correctly. I will retain the trigonometric values for angle A but seek a more direct path to the solution, avoiding this semiperimeter dead end.

### 3. Synthesizing a novel, correct solution from useful parts

*[Meta comment: After systematically identifying the flaws in Candidates 1, 3, and 4, and the dead-end in Candidate 2's approach, the model determines that a novel solution path is required. The chosen strategy is a synthesis: it leverages the correct trigonometric values for angle A, which were validated from Candidate 2's work, and integrates them with a powerful trigonometric identity not utilized by any of the candidates.]*

...We have $r = 6$, $R = 13$, so: $6 = 4 \cdot 13 \cdot \sin(A/2)\sin(B/2)\sin(C/2)$. We already found $\sin(A/2) = \sqrt{39}/13$. Let me plug that in: $6 = 52 \cdot (\sqrt{39}/13) \cdot \sin(B/2)\sin(C/2) \rightarrow 6 = 4\sqrt{39} \cdot \sin(B/2)\sin(C/2) \rightarrow \sin(B/2)\sin(C/2) = 6/(4\sqrt{39}) = 3/(2\sqrt{39})$. ... Also, since $B+C = \pi - A$, $\sin(B/2)\sin(C/2) = [\cos((B-C)/2) - \cos((B+C)/2)]/2 = [\cos((B-C)/2) - \sin(A/2)]/2$. So, $[\cos((B-C)/2) - \sqrt{39}/13]/2 = 3/(2\sqrt{39}) \rightarrow \cos((B-C)/2) - \sqrt{39}/13 = 3/\sqrt{39} \rightarrow \cos((B-C)/2) = 3/\sqrt{39} + \sqrt{39}/13 = \cdots = 2\sqrt{39}/13$. ... Now, using the identity for $\sin B \sin C$: $\sin B \sin C = [\cos(B - C) - \cos(B + C)]/2$. We have $\cos(B - C) = 2\cos^2((B - C)/2) - 1 = 2 \cdot ((2\sqrt{39}/13)^2) - 1 = \cdots = 143/169$. And $\cos(B + C) = -\cos A = -7/13$. $\sin B \sin C = [143/169 - (-7/13)]/2 = [143/169 + 91/169]/2 = (234/169)/2 = 117/169$. ... Thus, $AB \cdot AC = 4R^2 \sin B \sin C = 4 \cdot (13)^2 \cdot (117/169) = 4 \cdot 169 \cdot (117/169) = 4 \cdot 117 = 468$.

### 4. Final verification of the self-derived answer

...Hmm, so the correct answer should be 468. But none of the candidates have this. ... let me see if that makes sense with the area. The area $\Delta = (AB \cdot AC \sin A)/2 = (468 \cdot (2\sqrt{30}/13))/2 = (468 \cdot \sqrt{30})/13 = 36\sqrt{30}$. Also, the area is $rs = 6s$. Thus, $6s = 36\sqrt{30} \rightarrow s = 6\sqrt{30}$. ... The sides are $a = 4\sqrt{30}$, and $b + c = 2s - a = 12\sqrt{30} - 4\sqrt{30} = 8\sqrt{30}$. Thus, $b + c = 8\sqrt{30}$, and $bc = 468$. ... We can check if these satisfy the Law of Cosines: $\cos A = (b^2 + c^2 - a^2)/(2bc)$. We have $\cos A = 7/13$. ... $b^2 + c^2 = (b + c)^2 - 2bc = (8\sqrt{30})^2 - 2(468) = 1920 - 936 = 984$. Plugging into Law of Cosines: $(984 - 480)/(2 \cdot 468) = 504/936 = 7/13$. This matches the earlier result. Thus, the calculations are consistent. Therefore, the correct answer should be 468.

### H.3 SUMMARY COMPONENT OF SELF-REFINEMENT

We now present the summary component generated by the model. This component serves as a concise distillation of the core content detailed in the preceding chain-of-thought analysis. It is

worthy that the model autonomously provides a novel solution path, distinct from all candidate responses, to arrive at the correct answer of **468**.

---

**Summary component of GSR-7B**

To solve the problem, we need to find the product $AB \cdot AC$ for triangle $ABC$ with circumradius $R = 13$, inradius $r = 6$, and the condition that $\overline{IA} \perp \overline{OI}$.

Key Steps:

1. **Distance Between Incenter and Circumcenter**: Using Euler's formula:

$$OI^2 = R(R - 2r) = 13(13 - 12) = 13 \implies OI = \sqrt{13}$$

2. **Right Triangle Condition**: Since $\overline{IA} \perp \overline{OI}$, triangle $OIA$ is right-angled at $I$. Using the Pythagorean theorem:

$$OA^2 = OI^2 + IA^2 \implies 13^2 = (\sqrt{13})^2 + IA^2 \implies IA^2 = 156 \implies IA = \sqrt{156}$$

3. **Trigonometric Relationships**: - The distance $IA$ can also be expressed as $IA = 2R\sin\left(\frac{A}{2}\right)$:

$$\sqrt{156} = 2 \cdot 13 \cdot \sin\left(\frac{A}{2}\right) \implies \sin\left(\frac{A}{2}\right) = \frac{\sqrt{39}}{13}$$

- Using the double-angle identity for $\sin A$:

$$\sin A = 2\sin\left(\frac{A}{2}\right)\cos\left(\frac{A}{2}\right) = 2 \cdot \frac{\sqrt{39}}{13} \cdot \sqrt{\frac{130}{169}} = \frac{2\sqrt{30}}{13}$$

- The cosine of angle $A$ is:

$$\cos A = \sqrt{1 - \sin^2 A} = \frac{7}{13}$$

4. **Product of Sides Using Law of Sines**: - The sides $AB$ and $AC$ are given by:

$$AB = 2R\sin C \quad \text{and} \quad AC = 2R\sin B$$

- The product $AB \cdot AC$ is:

$$AB \cdot AC = 4R^2 \sin B \sin C$$

- Using trigonometric identities and the given conditions, we find:

$$\sin B \sin C = \frac{117}{169}$$

- Substituting $R = 13$ and $\sin B \sin C$:

$$AB \cdot AC = 4 \cdot 13^2 \cdot \frac{117}{169} = 4 \cdot 117 = 468$$

Final Answer:

$$\boxed{468}$$

---

