# OpenReview forum: "Learning to Refine: Self-Refinement of Parallel Reasoning in LLMs"
_ICLR.cc/2026/Conference — ICLR 2026 Conference Withdrawn Submission_

### Official Review · Reviewer_9xi4 · 2025-10-27

**Soundness:** 3
**Presentation:** 2
**Contribution:** 3
**Rating:** 4
**Confidence:** 3

**Summary:**

This paper introduce GSR, a novel approach that trains a model to aggregate multiple response generated in parallel by training with a hybrid training pipeline with a teacher model and a student model. The resulted model, GSR-7B, is evaluated on 5 different challenging math benchmarks. A detailed discussion is provided to show the robustness of the approach.

**Strengths:**

- The paper is overall easy-to-follow. The approach, while relatively simple, is very intuitive and reasonable.
- The experiments are comprehensive over multiple challenging math datasets. The baselines include multiple different reward models, and the detailed discussion shows the generalization capability of the proposed method.
- Compute budget are considered by reporting maj@5.

**Weaknesses:**

- Given GSR is a training-based algorithm, it is unclear whether directly training the underlying model on the same dataset with the same compute budget can lead to the same benefit.
- I find the idea of self-aggregation is relevant to a few recent work [1, 2, 3, 4]. While [4] is relatively new, I think [1, 2, 3] should be discussed and included as additional baselines.
- While in Sec. 4.5, additional results on different models are provided, those results are still after training. It is unclear whether the trained model, e.g., GSR-7B, can be directly combined with Qwen2.5-14B-Instruct.
- The models used are relatively old. I would appreciate it if the authors can include some more recent models released in this year, e.g., Deepseek-R1 series, or Qwen3 series, or GPT-oss series which include a more solid RL pipeilne during training to show the effectiveness of the approach on SOTA reasoning models.

Minor typo: In table 9, it probably should be MATH500, not MAHT500.

[1]. Yang X, An Y, Liu H, et al. Multiverse: Your Language Models Secretly Decide How to Parallelize and Merge Generation[J]. arXiv preprint arXiv:2506.09991, 2025.

[2]. Li Z, Feng X, Cai Y, et al. Llms can generate a better answer by aggregating their own responses[J]. arXiv preprint arXiv:2503.04104, 2025.

[3]. Yang C, Wang X, Lu Y, et al. Large language models as optimizers[C]//The Twelfth International Conference on Learning Representations. 2023.

[4]. Zhao W, Aggarwal P, Saha S, et al. The majority is not always right: Rl training for solution aggregation[J]. arXiv preprint arXiv:2509.06870, 2025.

**Questions:**

Please refer to the weakness section for general concerns. Additionally, here are some questions that involves clarity:
- Is the selfRef@4 in Sec. 4.3 on GSR or on the original model? Can you report the results of the alternative setting?
- In table 1, the caption says additional Pass@1 from  Yang et al. (2025c) and QwenTeam (2024) is reported, but I cannot find them anywhere.

---

> ### Author Response · Authors · 2025-11-28
> **Response to Reviewer 9xi4 (1/2)**
>
> > `Weakness 1: The efficacy of GSR training pipeline`
>
> We thank the reviewer for this important question. To validate the efficacy of our training method, we include an ablation study compared with a baseline of direct SFT in Sec 4.4 on the same data and the same training budget.
>
> *Table 1: Comparison of our hybrid training pipeline against pure distillation baseline (direct SFT)*
> | **Method**   | **Benchmark** | **Base Model** | **SFT (20k direct)** | **SFT (20k hybrid)** |
> |--------------|---------------|----------------|----------------------|----------------------|
> | **pass@1**   | AIME24        | 13.2           | **27.9**             | 27.5                 |
> |              | AIME25        | 7.2            | **26.7**             | 25.6                 |
> | **selfRef@4**| AIME24        | 15.6           | 37.5                 | **45.6**             |
> |              | AIME25        | 9.6            | 30.2                 | **32.7**             |
>
> As shown in the table 1, direct SFT method does not achieve the same gains within self-refinement as our full framework, indicating that the improvement is not explained by additional training alone.
>
> We find that direct SFT on refinement tasks is often insufficient to promote self-refinement behaviors, and the specialized training framework for structured refinement objectives matters.
>
> > `Weakness 2: The comparison with recent works`
>
> We thank the reviewer for pointing some recent works and agree that they should be discussed further. **The core contribution** of our paper is the design of a novel framework which enhances small scale model's self-refinement capability and thus surpasses other parallel test-time scaling baselines.
>
> Multiverse [1] is a new non-autoregressive generative model, internalizing a MapReduce paradigm to enable native parallel generation. However, it's **not a refinement framework**.
> GSA [2] is a prompting engineering techinique to synthesize an improved response from multiple samples **without any training pipeline**. However, as shown in Table 2, we find that for a small scale LLM, it's hard to simply prompt it to produce a sufficient improved response.
> LLMs as Optimizers (OPRO) [3], which use LLMs to **optimize prompts and instructions** rather than aggregating candidate responses for reasoning tasks.
> AggLM [4] trains a model to **aggregate model's own responses via RLVR**.
>
> Conceptually, these works focus on **architectural parallelization or prompt optimization or test-time aggregation on their own responses via prompting or RL**. However, our method trains a smaller student model **via distillation-refinement with dual objectives**, direct solving and parallel refinement, aiming to transfer both domain capability and refinement behavior from a stronger teacher.
>
> [1] Multiverse: Your Language Models Secretly Decide How to Parallelize and Merge Generation. arXiv:2506.09991.
>
> [2] LLMs Can Generate a Better Answer by Aggregating Their Own Responses. arXiv:2503.04104.
>
> [3] Large language models as optimizers. ICML 2023.
>
> [4] The majority is not always right: RL training for solution aggregation. arXiv:2509.06870.
>
> > `Weakness 3: Combination GSR-7B with other models`
>
> Thank you for pointing out the combination of GSR-7B with other models. Besides self-refinement (a model refines its own responses), GSR-7B **also support cross model refinement**, where GSR-7B acts as a refiner over responses generated by other policy models.
>
> We conduct an additional experiment using GSR-7B to refine responses from both a non-reasoning model (Qwen-2.5-32B-Instruct) and a reasoning model (DeepSeek-R1-Distill-Qwen-7B).
>
> *Table 2: Cross model refinement (Ref@4) using GSR-7B as a refiner against self-refinement (selfRef@4).*
> | **Policy Model**         | **AIME24** pass@1 | **AIME24** selfRef@4 | **AIME24** Ref@4(GSR-7B) | **AIME25** pass@1 | **AIME25** selfRef@4 | **AIME25** Ref@4(GSR-7B) |
> |--------------------------|-------------------|----------------------|--------------------------|-------------------|----------------------|--------------------------|
> | Qwen-2.5-32B-Instruct    | 16.7              | 22.3                 | **53.7** (+31.4)         | 12.9              | 16.3                 | **35.8** (+19.5)         |
> | R1-7B                    | 55.4              | 66.7                 | **74.6** (+7.9)          | 38.3              | 43.5                 | **52.9** (+9.4)          |
>
> As shown in Table 2, when refining responses from Qwen2.5-32B-Instruct, the final accuracy reaches 53.7 on AIME24 and 35.8 on AIME25. And GSR-7B reaches **74.6 (AIME24) and 52.9 (AIME25)** when refining responses from Deepseek-R1-Distill-Qwen-7B. These results demonstrate that GSR-7B generalizes beyond its own candidates responses and can be directly combined with other policy models.

---

> ### Author Response · Authors · 2025-11-28
> **Response to Reviewer 9xi4 (2/2)**
>
> > `Weakness 4: Add more recent models in the self-refinement settings`
>
> We additionally report the refinement gain ($\Delta$) defined as self-refinement gain over majority voting in Table 3 across both reasoning and non-reasoning model families ranging from 1.5B to 72B.
>
> *Table 3: Self-Refinement Gain over Majority Voting ($\Delta$) on AIME24 and AIME25 across reasoning and non-reasoning models raning from 1.5B to 72B.*
> | **Model**                        | **AIME24** pass@1 | **AIME24** maj@4 | **AIME24** selfRef@4 | **AIME24** $\Delta$ | **AIME25** pass@1 | **AIME25** maj@4 | **AIME25** selfRef@4 | **AIME25** $\Delta$ | **Avg. ($\Delta$)** |
> |-----------------------------------|-------------------|------------------|----------------------|---------------------|-------------------|------------------|----------------------|---------------------|---------------------|
> | Qwen-2.5-7B-Instruct             | 13.2              | 16.7             | 15.6                 | -1.1                | 7.2               | 9.6              | 11.3                 | 1.7                 | 0.3                 |
> | Qwen-2.5-14B-Instruct            | 13.5              | 17.5             | 16.0                 | -1.5                | 12.7              | 15.8             | 15.8                 | 0.0                 | -0.75               |
> | Qwen-2.5-32B-Instruct            | 16.7              | 19.2             | 22.3                 | 3.1                 | 12.9              | 15.8             | 16.3                 | 0.5                 | 1.8                 |
> | Qwen-2.5-72B-Instruct            | 17.3              | 20.0             | 22.1                 | 2.1                 | 10.6              | 11.7             | 13.3                 | 1.6                 | 1.85                |
> | **DeepSeek-R1-Distill-Qwen-1.5B**| 31.0              | 36.7             | 33.3                 | -3.4                | 24.0              | 27.5             | 24.8                 | -2.7                | -3.05               |
> | **DeepSeek-R1-Distill-Qwen-7B**  | 55.4              | 67.5             | 66.7                 | -0.8                | 38.3              | 46.7             | 43.5                 | -3.2                | -2.0                |
> | **DeepSeek-R1-Distill-Qwen-14B** | 67.1              | 77.5             | 78.3                 | 0.8                 | 48.5              | 58.3             | 58.9                 | 0.6                 | 0.7                 |
> | **QwQ-32B**                       | 79.6              | 85.0             | 85.4                 | 0.4                 | 67.1              | 75.0             | 76.9                 | 1.9                 | 1.15                |
>
>
> We observe that **the refinement gain is largely dependent with model scale rather than model's domain capability**. Specifically, this trend holds for both reasoning and non-reasoning model families: larger models (32B–72B) consistently achieve higher positive refinement gain than smaller models (1.5B–14B).
> Moreover, the refinement gain does not simply reflect domain capability. For example, DeepSeek-R1-Distill-Qwen-7B attains a much higher pass@1 than Qwen2.5-32B-Instruct (55.4 vs 16.7 on AIME24), yet its refinement gain is still negative (-2.0 vs 1.8 on average).
>
> This observation motivates our distillation-refinement framework. We use a large teacher model to **transfer both (i) domain competence and (ii) the refinement capability into a smaller student model**.
>
> > `Question 1: Is the selfRef@4 in Sec. 4.3 on GSR or on the original model`
>
> In our paper, self-refinement refers to the setting where the same model both generates its own candidate responses and refines them, consistent with the standard definition of self-refinement in prior work [5].
>
> Accordingly, the selfRef@4 results in Sec. 4.3 are evaluated on our model (GSR-7B), where we first sample four candidates from GSR-7B itself, and then let GSR-7B refine its own candidates.
>
> [5] Self-Refine: Iterative Refinement with Self-Feedback. NeurIPS 2023.
>
> > `Question 2: The results of Table 1 miss`
>
> Thank you for pointing out this. Concretely, the AIME24 and MATH500 results for larger model are copied verbatim from publicly reported results [6], while the AIME25 results are evaluated by our evaluation pipeline. We will revise the caption in late version.
>
> [6] ReasonFlux: Hierarchical LLM Reasoning via Scaling Thought Templates. NeurIPS 2025.

---

### Official Review · Reviewer_3N4x · 2025-10-28

**Soundness:** 2
**Presentation:** 2
**Contribution:** 2
**Rating:** 2
**Confidence:** 4

**Summary:**

This paper proposes Generative Self-Refinement (GSR), a parallel test-time scaling method that enables a single LLM to generate multiple candidate solutions and then self-refine them into a superior answer. A hybrid training pipeline jointly optimizes direct-solving and refinement via teacher-student distillation on math data.

**Strengths:**

1. The paper is written clearly;
2. The overall training pipeline is relatively simple.

**Weaknesses:**

1. The paper explores Self-Refinement but lacks detailed discussion of closely related works such as SELF-REFINE[1], SCORE [2], ARIES [3], ReVISE [4] and SynPO [5]：
   * SELF-REFINE also focuses on refining a model’s own outputs—what distinguishes GSR from SELF-REFINE?
   * SCORE investigates self-correction in mathematical reasoning—what is the conceptual difference between self-refinement and self-correction?
   * ARIES similarly studies self-refinement, and GSR’s SFT loss function appears quite similar to that of ARIES. What are the key distinctions between GSR and ARIES?
   * ReVISE and SynPO also explores refinement through verification or refiner-based mechanisms, yet the paper provides little discussion of these relevant studies.
2. The proposed approach essentially distills a smaller model from a stronger teacher model, which makes it highly dependent on teacher quality. The paper lacks analysis of this dependency.
3. Figure 4 shows that as the number of candidates increases, the performance of Self-Refinement becomes worse than Majority Voting. This suggests that the model’s self-refinement ability may be largely forced by distillation; once the number of candidates deviates from that seen during training, performance stagnates or even degrades, indicating poor generalization over candidate count.
4. The authors could further explore the model’s refinement behavior, for example:
   - Is Self-Refinement always beneficial, or does it sometimes turn correct answers into incorrect ones? What are the respective proportions of “correct → wrong” and “wrong → correct”?
   - Can the proposed strategy perform iterative self-refinement—refining its own answers multiple times—or is it limited to a single refinement pass over the candidates?
5. As a decoding strategy, the paper lacks fine-grained ablation studies, such as analyzing how parameters like temperature and top-p affect the performance of Majority Voting and Self-Refinement.
6. The paper also lacks direct comparisons with related methods such as SCORE [2], ARIES [3], and ReVISE [4].



[1] Self-Refine: Iterative Refinement with Self-Feedback. NeurIPS 2023.

[2] Training Language Models to Self-Correct via Reinforcement Learning. ICLR 2025.

[3] ARIES: Stimulating Self-Refinement of Large Language Models with and for Iterative Preference Optimization. ICLR 2025 Workshop.

[4] ReVISE: Learning to Refine at Test-Time via Intrinsic Self-Verification. ICML 2025.

[5] Self-Boosting Large Language Models with Synthetic Preference Data ICLR 2025.

**Questions:**

Please refer to Weaknesses.

---

> ### Author Response · Authors · 2025-11-28
> **Response to Reviewer 3N4x (1/3)**
>
> > `Weakness 1: Lack detailed discussion of closely related works`
>
> Thank you for pointing out the omission of detailed discussion of closely related approaches [1,2,3,4,5]. We agree that a clearer differentiation is required. The core contribution of our paper is the design of a novel framework which enhances small scale model's self-refinement capability and thus surpasses other parallel test-time scaling baselines.
>
> Self-Refine[1] uses a single model to generate an initial ouput and use feedback to refine it iteratively at inference time. It's a iterative self-refinement method (**not parallel**) and does not involve any training pipeline.
> ReVISE[4] trains a model via a two‑stage curriculum to perform self‑verification and then self‑correction during inference (using a special [refine] token and candidate verification).
> SCoRe[2] employs a online reinforcement learning framework to train an LLM  improve self‑correction in inference time.
> SynPO[5] adopts synthetic preference data for self‑boosting.
> ARIES[3] leverages iterative preference training and inference‑time refinement.
>
> These methods basiclly train a model to refine their own outputs through rejection sampling, preference training, RL and etc. They are all **self-refinement framework during training**, which models either refine their own outputs at inference, or are trained for self-improvement via feedback or verification loops. Besides, all these methods are **sequential iterative refinement**, not **parallel aggregation refinement**.
>
> In contrast, our method is a **distillation refinement framework during training**. We first distil a large teacher model into a smaller student model, and then train the student with two parallel objectives, direct solving and refinement. By doing so, the student model not only acquires domain capability, but also works well on refining outputs.
>
> We will incorporate these distincitions in the late revised manuscript.
>
> [1] Self-Refine: Iterative Refinement with Self-Feedback. NeurIPS 2023.
>
> [2] Training Language Models to Self-Correct via Reinforcement Learning. ICLR 2025.
>
> [3] ARIES: Stimulating Self-Refinement of Large Language Models with and for Iterative Preference Optimization. ICLR 2025 Workshop.
>
> [4] ReVISE: Learning to Refine at Test-Time via Intrinsic Self-Verification. ICML 2025.
>
> [5] Self-Boosting Large Language Models with Synthetic Preference Data. ICLR 2025.

---

> ### Author Response · Authors · 2025-11-28
> **Response to Reviewer 3N4x (2/3)**
>
> > `Weakness 2: Lacks analysis of teacher model's quality`
>
> Thank you for pointing out the dependency of our approach on a strong teacher model. Indeed, this dependency is by design rather than an unintended limitation. We deliberately use a large scale model to enable the student model not only to inherit domain‑capability but also to **learn a self‑improvement mechanism**.
>
> To support this, we have added a new experimental table (Table 1), which shows the gain from self‑improvement against majority voting correlates strongly with model scale but shows no strong correlation with domain capability. This finding empirically to support our design, using a large scale teacher model to transfer its **self‑improvement capability that small models would otherwise lack**.
>
> *Table 1: Self-Refinement Gain over Majority Voting ($\Delta$) on AIME24 and AIME25 across reasoning and non-reasoning models raning from 1.5B to 72B*
> | **Model**                        | **AIME24** pass@1 | **AIME24** maj@4 | **AIME24** selfRef@4 | **AIME24** $\Delta$ | **AIME25** pass@1 | **AIME25** maj@4 | **AIME25** selfRef@4 | **AIME25** $\Delta$ | **Avg. ($\Delta$)** |
> |-----------------------------------|-------------------|------------------|----------------------|---------------------|-------------------|------------------|----------------------|---------------------|---------------------|
> | Qwen-2.5-7B-Instruct             | 13.2              | 16.7             | 15.6                 | -1.1                | 7.2               | 9.6              | 11.3                 | 1.7                 | 0.3                 |
> | Qwen-2.5-14B-Instruct            | 13.5              | 17.5             | 16.0                 | -1.5                | 12.7              | 15.8             | 15.8                 | 0.0                 | -0.75               |
> | Qwen-2.5-32B-Instruct            | 16.7              | 19.2             | 22.3                 | 3.1                 | 12.9              | 15.8             | 16.3                 | 0.5                 | 1.8                 |
> | Qwen-2.5-72B-Instruct            | 17.3              | 20.0             | 22.1                 | 2.1                 | 10.6              | 11.7             | 13.3                 | 1.6                 | 1.85                |
> | **DeepSeek-R1-Distill-Qwen-1.5B**| 31.0              | 36.7             | 33.3                 | -3.4                | 24.0              | 27.5             | 24.8                 | -2.7                | -3.05               |
> | **DeepSeek-R1-Distill-Qwen-7B**  | 55.4              | 67.5             | 66.7                 | -0.8                | 38.3              | 46.7             | 43.5                 | -3.2                | -2.0                |
> | **DeepSeek-R1-Distill-Qwen-14B** | 67.1              | 77.5             | 78.3                 | 0.8                 | 48.5              | 58.3             | 58.9                 | 0.6                 | 0.7                 |
> | **QwQ-32B**                       | 79.6              | 85.0             | 85.4                 | 0.4                 | 67.1              | 75.0             | 76.9                 | 1.9                 | 1.15                |
>
> > `Weakness 3: Performance of self-refinement as the number of candidates increases`
>
>
> Thank you for your thorough review! After reviewing our results, we would like to point out the following:
>
> Our model is trained at a fixed candidate pool size of four. We observe that when the number of candidate responses increases moderately (e.g. up to 10), the performance of self-refinement continues to improve and outperforms majority voting. That demonstrates that the student model after distillation exhibits **significant extrapolation capability** beyond the candidate count seen during training.
>
> The performance begins to degrade only when the candidate pool becomes very large. We posit that this degradation is not due to the design of our method, but rather stems from a more general limitation of LLMs under very long input contexts, which is recently documented as context rot [6].
>
> [6] Context Length Alone Hurts LLM Performance Despite Perfect Retrieval. arXiv:2510.05381.

---

> ### Author Response · Authors · 2025-11-28
> **Response to Reviewer 3N4x (3/3)**
>
> > `Weakness 4: Further exploration about self-refinement behavior`
>
> **SubQuestion 1: Is Self-Refinement always beneficial?**
>
> From Table 1, we can find empirically across model families and scales on five challenging mathematical benchmarks, self-refinement consistently improves over pass@1, which indicates **the benefit is not restricted to a particular domain capability or a single model type**.
>
> To further understand how self-refinement works, we counts average accuracy rate by the number of correct candidate responses (0-4) in the pool of 4 (Table 2). When the candidates has high consensus (3–4 correct candidates), self-refinement achieves near-ceiling performance (95–100\%). In contrast, under low consensus (0–1 correct candidate), our method still recover wrong candidates to correct (e.g., 49–66\% success when only one candidate is correct), and **it can even recover correct answers in the all-wrong setting**.
>
> *Table 2: Average accuracy rate by the number of correct candidates(0-4) in the pool of 4 across different benchmarks*
> | **Benchmark** | **4**   | **3**   | **2**   | **1**   | **0**   |
> |---------------|---------|---------|---------|---------|---------|
> | AIME24        | 100     | 97.4    | 89.6    | 60.2    | 5.9     |
> | AIME25        | 100     | 95.8    | 87.1    | 65.6    | 4.5     |
> | AMC22-23      | 99.9    | 99.3    | 80.0    | 53.6    | 9.0     |
> | MATH500       | 99.8    | 94.3    | 76.4    | 49.1    | 3.4     |
> | Olympiad      | 99.8    | 94.0    | 81.0    | 50.7    | 4.4     |
>
>
> **SubQuestion 2: Can the proposed strategy perform iterative self-refinement?**
>
> Yes. While our method is designed as a parallel aggregation over a candidate pool for compute efficiency and stability, it can be **naturally extended to sequential iterative refinement procedure**. We can generate a candidate pool parallelly, refine some responses from the pool and then feed the refined solutions back to constitute a new pool. The recent study [7] empirically shows that repeated refinement and synthesis procedures are feasible and outperform baselines such as majority voting.
>
> [7] CoT-based Synthesizer: Enhancing LLM Performance through Answer Synthesis. ACL 2025.
>
> > `Weakness 5: Lacks fine-grained ablation studies about temperature or top-p`
>
> Following recent best practices in reproducibility for reasoning benchmarks [8], We use temperature=0.6 and top-p=0.95, report results averaged over 16–32 independent runs, and keep the decoding configuration fixed across all methods to ensure a fair comparison.
>
> We acknowledge that a more fine-grained ablation study about temperature/top-p is necessary. We will add in the late version.
>
> [8] A Sober Look at Progress in Language Model Reasoning: Pitfalls and Paths to Reproducibility. COLM 2025.
>
> > `Weakness 6: Lacks direct comparisons with related methods such as SCORE, ARIES, and ReVISE`
>
> We appreciate the suggestion to include direct baselines such as SCoRe, ARIES, ReVISE. **The core contribution** of our paper is the design of a novel framework which enhances small scale model's self-refinement capability and thus surpasses other parallel test-time scaling baselines.
> However, we find that these methods are not straightforward to reproduce or directly compare in our experimental setting.
>
> These methods all primarily adopt **a sequential and multi-turn refinement**, which incurs substantially different inference-time cost and often longer contexts.
> To be concrete, SCoRe is a multi-turn online RL approach for self-correction. ReVISE trains a model via a two-stage curriculum to perform self-verification and then self-correction during inference. ARIES cultivates self-refinement through iterative preference training and leverages it during inference.
>
> In contrast, our method instead focuses on **one-pass parallel aggregation refinement**. Therefore, a direct comparison is only meaningful under matched inference budgets (number of turns, total tokens/context length). This mismatch of settings makes a baseline comparison potentially misleading.
>
> We will add these discussions in the revised late version.

---

### Official Review · Reviewer_mw5D · 2025-10-31

**Soundness:** 4
**Presentation:** 3
**Contribution:** 3
**Rating:** 6
**Confidence:** 4

**Summary:**

This paper introduces Generative Self-Refinement (GSR), a novel parallel test-time scaling framework designed to enhance Large Language Models (LLMs) in solving complex, multi-step reasoning problems. Unlike existing test-time scaling methods such as Best-of-N or majority voting, which are limited by the quality of candidate responses and cannot produce a correct solution if all candidates are flawed, GSR empowers a single, unified LLM to both generate diverse candidate responses in parallel and then perform self-refinement to synthesize a superior solution. This refinement is achieved by prompting the model with the original problem and its self-generated candidates, enabling it to diagnose flaws and selectively leverage insights, even from erroneous solutions, to construct a correct answer.

Overall, this paper shows a promising way of teaching the model to do self-refine for test-time scaling instead of designing complex pipelines, which could be a valuable insight and paradigm for the community.

**Strengths:**

1.  **Significant Conceptual Novelty in Self-Correction:** The core strength of GSR lies in its potential to **correct solutions even when all initial candidates are flawed**, representing a significant conceptual advance over selection-based methods like majority voting. This self-correction mechanism pushes the boundary of model capability and introduces a promising new paradigm for internal consistency checking.

2.  **Clarity and Technical Soundness:** The motivation for the Generative Self-Refinement method is clearly articulated. The proposed methodology is technically sound, and the paper is well-structured and easy to follow, making the complex topic accessible.

3.  **Extensive Experimental Validation:** The experimental validation is extensive and comprehensive. The authors thoroughly investigate the impact of the hybrid Supervised Fine-Tuning (SFT) dataset on the model's refinement ability, demonstrate the robust scaling benefits across different model sizes, and analyze the accuracy evolution as the number of candidates is increased. This thoroughness is commendable.

**Weaknesses:**

- **Baseline selection need to be improved:** The primary weakness lies in the authors compare the GSR with pure test-time scaling (TTS) methods as baseline, which is a bit unfair. Given that **GSR requires dedicated Supervised Fine-Tuning (SFT) on a specialized dataset** (collected following Section 3.3) to unlock the self-refinement skill, it operates as a _training technique_ rather than a generalized, plug-and-play TTS method. This distinction is critical and should be clarified. Furthermore, the experimental results (e.g., Qwen2.5-7B-Instruct on several mathematical benchmarks) show that the proposed method (`selfRef@4`) often performs comparably to or _worse_ than the simpler baseline of majority voting (`maj@4`) without the proposed SFT process. This lack of consistent, significant uplift, combined with the prerequisite training, undermines its positioning as a superior TTS method. The authors must either justify this classification or **compare GSR against other methods that explicitly train internal reasoning or reflection abilities** (e.g., Chain-of-Thought approaches with explicit verification steps).
- Lack of Sensitivity Analysis to Candidate Diversity and Quality: The efficacy of both the self-refinement and majority voting components is inherently tied to the **quality and diversity of the initial candidate pool**. The current analysis lacks a systematic investigation into the sensitivity of GSR performance against these critical metrics. A thorough ablation study is warranted to explore how GSR behaves when candidates exhibit low diversity (i.e., multiple identical, but incorrect, responses) or low aggregate quality. Providing stronger evidence for its robustness beyond a simple majority selection process is necessary.

### Minor Comments
- Subscripts should be wrapped in \text{} when it is not a variable. For example, $\mathcal{D}_{\text{direct}}$ and $\mathcal{D}_{\text{selfR}}$ instead of $\mathcal{D}_{direct}$ and $\mathcal{D}_{selfR}$.

**Questions:**

- How do you guarantee that the teacher model has the ability of drawning correct answers from the candidates proposed by the students?
- The accuracy saturates as the number of candidate increases. This contradicts the goal of test-time "scaling" methods, where we expect performance gains from increased computational resources. Have the authors investigated the scalability of the Supervised Fine-Tuning (SFT) dataset size?

---

> ### Author Response · Authors · 2025-11-28
> **Response to Reviewer mw5D(1/2)**
>
> >`Weakness 1 (part I): Unfair comparison with parallel test time scaling methods`
>
> Thank you for pointing out the distinction. We agree that our method is training-based and should not be defined as a pure plug-and-play and training-free parallel TTS trick. We will revise the draft to make this more explicit and precise.
>
> However, comparing to TTS baselines is **still necessary and fair** because our evaluation settings focus on the same test-time scaling procedure: we first sample a candidate response pool and aggregate **the same candidate responses** using all baselines. In this setting, Best-of-N and majority voting are widely adopted.
>
> Moreover, since training process improves the student model’s domain capability, all aggregation baselines (e.g., BoN/majority voting) **also benefit**. From the experiments, our key finding is that the learned refinement skill delivers a consistently larger additional gain over these baselines.
>
>
> >`Weakness 1 (part II): The classification of our methods`
>
> To motivate why our training pipeline is necessary, we add a systematic study across reasoning and non-reasoning model families from 1.5B to 72B (Table 1). We find that refinement gain (accuracy of self-refinement gain against majority voting) is **strongly dependent with model's scale rather than model's domain ability**. Smaller models often have near zero or even negative refinement gain while larger ones exhibit significantly positive refinement gain.
>
> Therefore, for small models, a dedicated training pipeline is required to reliably acquire this refinement ability.
>
> *Table1: Self-Refinement Gain over Majority Voting ($\Delta$) on AIME24 and AIME25 across reasoning and non-reasoning models ranging from 1.5B to 72B.*
> | **Model**                        | **AIME24** pass@1 | **AIME24** maj@4 | **AIME24** selfRef@4 | **AIME24** $\Delta$ | **AIME25** pass@1 | **AIME25** maj@4 | **AIME25** selfRef@4 | **AIME25** $\Delta$ | **Avg. ($\Delta$)** |
> |-----------------------------------|-------------------|------------------|----------------------|---------------------|-------------------|------------------|----------------------|---------------------|---------------------|
> | Qwen-2.5-7B-Instruct             | 13.2              | 16.7             | 15.6                 | -1.1                | 7.2               | 9.6              | 11.3                 | 1.7                 | 0.3                 |
> | Qwen-2.5-14B-Instruct            | 13.5              | 17.5             | 16.0                 | -1.5                | 12.7              | 15.8             | 15.8                 | 0.0                 | -0.75               |
> | Qwen-2.5-32B-Instruct            | 16.7              | 19.2             | 22.3                 | 3.1                 | 12.9              | 15.8             | 16.3                 | 0.5                 | 1.8                 |
> | Qwen-2.5-72B-Instruct            | 17.3              | 20.0             | 22.1                 | 2.1                 | 10.6              | 11.7             | 13.3                 | 1.6                 | 1.85                |
> | **DeepSeek-R1-Distill-Qwen-1.5B**| 31.0              | 36.7             | 33.3                 | -3.4                | 24.0              | 27.5             | 24.8                 | -2.7                | -3.05               |
> | **DeepSeek-R1-Distill-Qwen-7B**  | 55.4              | 67.5             | 66.7                 | -0.8                | 38.3              | 46.7             | 43.5                 | -3.2                | -2.0                |
> | **DeepSeek-R1-Distill-Qwen-14B** | 67.1              | 77.5             | 78.3                 | 0.8                 | 48.5              | 58.3             | 58.9                 | 0.6                 | 0.7                 |
> | **QwQ-32B**                       | 79.6              | 85.0             | 85.4                 | 0.4                 | 67.1              | 75.0             | 76.9                 | 1.9                 | 1.15                |
>
> In Sec 4.4, we further conduct a controlled ablation study where three SFT variants are trained on the same set of questions with the same data size (20K): (i) direct-only distillation, (ii) refinement-only training, and (iii) hybrid (10K+10K).
>
> Results demonstrate that pure distillation baseline learns a weak refinement ability and refinement only baseline is bottlenecked by lower pass@1 and thus a lower performace ceiling.
> Our training pipeline **attains the best overall refinement performance while preserving strong direct-solving ability**.

---

> > ### Author Response · Authors · 2025-11-28
> > **Response to Reviewer mw5D(2/2)**
> >
> > > `Weakness 2: Quality and diversity of the initial candidate response pool`
> >
> > Thank you for pointing out this concern.
> >
> > We count average accuracy rate by the number of correct candidate responses (0-4) in the pool of 4 (Table 2). When the candidates has high consensus (3–4 correct candidates), self-refinement achieves near-ceiling performance (95–100\%). In contrast, under low consensus (0–1 correct candidate), our method still **recover wrong candidates to correct** (e.g., 49–66\% success when only one candidate is correct), and it can **even recover correct answers in the all-wrong setting**.
> >
> > *Table 2: Average accuracy rate by the number of correct candidates(0-4) in the pool of 4 across different benchmarks*
> >
> > | **Benchmark** | **4**   | **3**   | **2**   | **1**   | **0**   |
> > |---------------|---------|---------|---------|---------|---------|
> > | AIME24        | 100     | 97.4    | 89.6    | 60.2    | 5.9     |
> > | AIME25        | 100     | 95.8    | 87.1    | 65.6    | 4.5     |
> > | AMC22-23      | 99.9    | 99.3    | 80.0    | 53.6    | 9.0     |
> > | MATH500       | 99.8    | 94.3    | 76.4    | 49.1    | 3.4     |
> > | Olympiad      | 99.8    | 94.0    | 81.0    | 50.7    | 4.4     |
> >
> >
> > Regarding candidate diversity, we will additionally report controlled sampling settings (e.g. temperature) to identify diversity effects under comparable pool quality in the late version.
> >
> > > `Question 1: How to guarantee teacher model to draw correct answers from candidates`
> >
> > As discussed in Appendix F.1, we adopt a **rejection sampling strategy**. After refining, we remove all incorrect answers which the teacher fails to recover from the candidates responses proposed by students.
> >
> >
> > > `Question 2: Have the authors investigated the scalability of SFT size`
> >
> > Thank you for your question about scalability of SFT size.
> > Due to compute and time constraints during the rebuttal period, we did not run a full scaling study over SFT dataset size. We will include this experiment in future work.

---

### Official Review · Reviewer_ookS · 2025-10-31

**Soundness:** 1
**Presentation:** 2
**Contribution:** 1
**Rating:** 2
**Confidence:** 4

**Summary:**

The paper introduces a parallel test time scaling framework labeled Generative Self-Refinement, which improves the reasoning of a large language model by creating parallel traces and iteratively improving them using a verification pipeline. This refinement approach, along with the proposed hybrid training solution, improves downstream reasoning benchmarks by significant margin.

**Strengths:**

- Strong empirical results from the base model, but likely due to knowledge distillation and not self refinement.

**Weaknesses:**

- The method introduced by the paper is defined as self-refinement, however the only self-refinement part is the first set in Section 3.2, where the model itself is prompted to become a critic of its own generations. Subsequently, the proposed approach turns into a distillation problem, where a teacher model is used to provide additional signal. I fail to understand how this can be deemed as "self-refinement".
- From the experiment details, it is clear the proposed approach is distillation of QwQ-32B model into Qwen2.5-7B-Instruct. However, the baselines does not reflect the performance of QwQ-32B (from Table 1). Interestingly, QwQ-32B significantly outperforms the proposed method on AIME24 (79.5 vs 66.0 reported in the paper)
- Therefore, all reported results can just be taken to be a distillation effort from QwQ-32B, leaving the proposed title and method misleading as the improvements are not coming from self-refinement.
- If the authors intend to treat this as a distillation work, then significant rewriting of the draft is required, along with pure distillation based baselines.

**Questions:**

- Please clarify where self-refinement is being useful in this setup.

---

> ### Author Response · Authors · 2025-11-28
> **Response to Reviewer ookS(1/2)**
>
> > `Weakness 1&3`: It is distillation rather than self-refinement
>
> We appreciate your concern and agree that current writing could be confusing.
> In our paper, **self-refinement** refers to a test-time scaling manner, where a model first generate a pool of candidate responses and then aggregate them into a superior response. The teacher model is **used only during training** to provide supervision. We don't use any teacher model **during test-time**.
>
> **The core contribution** of our paper is the design of a novel framework which enhances small scale model's self-refinement capability and thus surpasses other parallel test-time scaling baselines.
>
> To further clarify the motivation, we add a comprehensive study across both reasoning and non-reasoning model families ranging from 1.5B to 72B. As shown in Table 1, **the refinement gain (selfRef@4 - maj@4) increases with model scale and is weakly correlated to model's domain capability**. This suggests that refinement skill is a distinct capability that does not automatically emerge from stronger domain capability alone. Motivated by this observation, we propose distillation-refinement framework, using a large teacher model to transfer both (i) domain competence and (ii) the refinement capability into a smaller student model.
>
> *Table 1: Self-Refinement Gain over Majority Voting ($\Delta$) on AIME24 and AIME25 across reasoning and non-reasoning models raning from 1.5B to 72B*
> | **Model**                        | **AIME24** pass@1 | **AIME24** maj@4 | **AIME24** selfRef@4 | **AIME24** $\Delta$ | **AIME25** pass@1 | **AIME25** maj@4 | **AIME25** selfRef@4 | **AIME25** $\Delta$ | **Avg. ($\Delta$)** |
> |-----------------------------------|-------------------|------------------|----------------------|---------------------|-------------------|------------------|----------------------|---------------------|---------------------|
> | Qwen-2.5-7B-Instruct             | 13.2              | 16.7             | 15.6                 | -1.1                | 7.2               | 9.6              | 11.3                 | 1.7                 | 0.3                 |
> | Qwen-2.5-14B-Instruct            | 13.5              | 17.5             | 16.0                 | -1.5                | 12.7              | 15.8             | 15.8                 | 0.0                 | -0.75               |
> | Qwen-2.5-32B-Instruct            | 16.7              | 19.2             | 22.3                 | 3.1                 | 12.9              | 15.8             | 16.3                 | 0.5                 | 1.8                 |
> | Qwen-2.5-72B-Instruct            | 17.3              | 20.0             | 22.1                 | 2.1                 | 10.6              | 11.7             | 13.3                 | 1.6                 | 1.85                |
> | **DeepSeek-R1-Distill-Qwen-1.5B**| 31.0              | 36.7             | 33.3                 | -3.4                | 24.0              | 27.5             | 24.8                 | -2.7                | -3.05               |
> | **DeepSeek-R1-Distill-Qwen-7B**  | 55.4              | 67.5             | 66.7                 | -0.8                | 38.3              | 46.7             | 43.5                 | -3.2                | -2.0                |
> | **DeepSeek-R1-Distill-Qwen-14B** | 67.1              | 77.5             | 78.3                 | 0.8                 | 48.5              | 58.3             | 58.9                 | 0.6                 | 0.7                 |
> | **QwQ-32B**                       | 79.6              | 85.0             | 85.4                 | 0.4                 | 67.1              | 75.0             | 76.9                 | 1.9                 | 1.15                |
>
> Importantly, in our main experiments we evaluate under the self-refinement setting and find that the self-refinement results **consistently outperform all other aggregation baselines**, which support the efficacy of our framework.
>
>
>
> > `Weakness 2`: Comparsion with teacher model QwQ-32B
>
> We agree that the teacher QwQ-32B is stronger in absolute accuracy (e.g., 79.6 of pass@1 on AIME24). However, our goal is **not to surpass the teacher on the domain capability**. Our contribution is to transfer both domain capability and refinement ability from a large teacher to a much smaller student.
>
> As shown in Table 1, although QwQ-32B is much stronger in pass@1 (79.6/67.1 on AIME24/25), GSR-7B **achieves substantially larger refinement gains** (+6.0/+5.0 on AIME24/25) in Sec 4.2.
> This demonstrates that our method transfers a refinement ability rather than merely imitating the teacher’s final answers.

---

> > ### Author Response · Authors · 2025-11-28
> > **Response to Reviewer ookS(2/2)**
> >
> > > `Weakness 4`: Add pure distillation based baselines
> >
> > We thank the reviewer for this important question. To validate the efficacy of our training method, we include an ablation study from Sec 4.4 compared with a pure distillatoin baseline on the same data and the same training budget.
> >
> > *Table 2:  Comparison of our hybrid training pipeline against pure distillation baseline (direct)*
> >
> > | **Method**   | **Benchmark** | **Base Model** | **SFT (20k direct)** | **SFT (20k hybrid)** |
> > |--------------|---------------|----------------|----------------------|----------------------|
> > | **pass@1**   | AIME24        | 13.2           | **27.9**             | 27.5                 |
> > |              | AIME25        | 7.2            | **26.7**             | 25.6                 |
> > | **selfRef@4**| AIME24        | 15.6           | 37.5                 | **45.6**             |
> > |              | AIME25        | 9.6            | 30.2                 | **32.7**             |
> >
> >
> > As shown in the table 2, pure distillation method does not achieve the same gains within self-refinement as our full framework, indicating that the improvement is not explained by additional training alone.
> > We further find that model after pure distillation is often insufficient to promote self-refinement behaviors, and the specialized training framework for structured refinement objectives matters.
> >
> > > `Question`: Where self-refinement is being useful in this setup
> >
> > Self-refinement is useful **at test-time** (our main experiments), where it acts as a learned "refinement operator". In our setup, GSR-7B first generate multiple candidates responses and then aggregate these candidate responses to synthesize a final solution.

---

### Note · Authors · 2025-12-02

I have read and agree with the venue's withdrawal policy on behalf of myself and my co-authors.